



# A subgrid method for the linear inertial equations of a compound flood model

Maarten van Ormondt[1], Tim Leijnse[2,3], Roel de Goede[2], Kees Nederhoff[1], Ap van Dongeren[2,4]

[1] Deltares USA, 8601 Georgia Ave, Silver Spring, MD 20910, USA
[2] Marine and Coastal Management, Deltares, Boussinesqweg 1, Delft, 2629 HV, The Netherlands
[3] Institute for Environmental Studies (IVM), Vrije Universiteit Amsterdam, De Boelelaan 1111, 1081 HV Amsterdam, The Netherlands.
[4] IHE Delft, Water Science and Engineering Dept, Westvest 7, 2611 AX Delft, The Netherlands

*Correspondence to*: Kees Nederhoff (kees.nederhoff@deltares-usa.us)

**Keywords.** Hydrodynamic modeling, subgrid, Linear Inertial Equations, compound flooding, SFINCS

**Abstract.** Accurate flood risk assessments and early warning systems are needed to protect and prepare people in coastal areas from storms. In order to provide this information efficiently and on time, computational costs need to be kept as low as possible. Reduced-complexity models using linear inertial equations and subgrid approaches have been used previously to achieve this goal. In this paper, for the first time, we developed a subgrid approach for the Linear Inertial Equations (LIE) that account for bed level and friction variations. We implemented this method in the SFINCS model. Pre-processed lookup tables that correlate water levels with hydrodynamic quantities make more precise simulations with lower computational costs possible. These subgrid corrections have undergone validation through a variety of conceptual and real-world application scenarios, including analyses of hurricane hazards and tidal fluctuations. We demonstrate that the subgrid corrections for Linear Inertial Equations significantly improve model accuracy while utilizing the same resolution without subgrid corrections. Moreover, coarser model resolutions with subgrid corrections can provide the same accuracy as finer resolutions without subgrid corrections. Limitations are discussed, for example, when grids do not adequately resolve river meanders, fluxes can be overestimated. Our findings show that subgrid corrections are an invaluable asset for hydrodynamic modelers striving to achieve a balance between accuracy and efficiency.



## 1   Introduction

With hundreds of millions of people living in areas with an elevation of less than 10 meters above sea level (McGranahan et
al, 2007), coastal zone flooding has large consequences for casualties and damage to real estate and infrastructure. To protect
and mitigate flood damages and loss of life, a priori risk assessments may inform decision makers in what locations and under
what circumstances flooding occurs, and what interventions to take. Furthermore, flood early warning systems provide
information based on which evacuation of citizens can take place to save lives. Both the risk assessments and early warning
systems should provide as accurate as possible information so as not to give false warnings or needlessly over or underestimate
the extent and cost of interventions.
For flood warnings, this means that simple bathtub approaches, where a peak water level is imposed on an area's topography,
do not suffice. They may overestimate the flood intensity because the surge hydrograph is not taken into account (Vousdoukas
et al., 2016), or underestimate it due to lacking physics (e.g. wave effects, Didier et al., 2020) or lacking inputs such as
roughness effects which would impede flow (Ramirez et al., 2016). Therefore, for a more accurate flood estimate, the dynamic
aspects of floods such as the duration of an event, and the path that flood waters take should be considered. Furthermore, the
compound nature of coastal area floods, which may be caused by marine surges, wave overtopping, coastal river discharges,
and local rainfall needs to be taken into account. These dynamics and processes may be resolved using process-based numerical
models which are based on the conservation of mass and momentum. However, classical full-physics models (ADCIRC;
Luettich et al., 1992, Delft3D-FLOW; Lesser et al., 2004, MIKE; Warren and Bach, 1992 or SOBEK; Stelling et al., 1998) are
computationally expensive, which limits their application for large areas and high resolution, and the exploration of
uncertainties in flooding due to uncertain inputs.

To that end, reduced-complexity models have been developed and applied in riverine settings and coastal applications.
Examples include, among others, the LISFLOOD(-FP) model by Bates et al. (2010 and the SFINCS (Super-Fast INundation
of CoastS) model by Leijnse et al. (2021)).  These models solve only the essential terms in the momentum equations using a
simple numerical scheme and are as a consequence orders of magnitude faster than the conventional models. Still, the number
of simulations that can be run is limited, as the numerical scheme is explicit and therefore strongly influenced by the spatial
grid size (and associated time step).

One way to further increase the computational speed is to apply a subgrid approach which makes use of the assumption that
water level gradients are typically much smaller than topographic gradients. Defina (2000) presented shallow water equations
with mass conservation corrections to account for wetting and drying areas, and corrections to the momentum equations to
account for varying velocities. Casulli (2009) introduced a dual-grid approach with a higher resolution grid for the bathymetry
and a lower resolution grid for the hydrodynamics where the depth and cross-sectional area were computed using the higher-



resolution grid and stored in lookup tables which were used to evaluate the water levels on the lower resolution grid. Volp et
al. (2013) extended Casulli's approach to finite volumes and incorporated a subgrid-based method to compute advection and
bottom friction under the assumptions of locally uniform flow direction and friction slope.  Sehili et al. (2014) showed that a
subgrid approach could save an order of magnitude of computational cost without major accuracy loss in estuarine modeling.
For coastal storm surge applications, Kennedy et al. (2019) developed a refined set of equations incorporating extra terms
derived from an upscaling technique. These additional terms, emerging from the averaging of shallow water equations, account
for the integral properties of fine-scale bathymetry, topography, and flow dynamics. This process is similar to how Boussinesq
approximations are used for turbulence closure in Navier-Stokes models and involves using coarse-scale variables, such as
averaged fluid velocity, to represent these fine-scale integrals.   They showed the improved performance of their model for the
case of tidal flooding in a small bay. Woodruff et al. (2021) extended this analysis to a case of storm surge with realistic
atmospheric forcing and reported a speedup of ADCIRC with a factor of 10-50. Similarly, Begmohammadi et al. (2023)
adapted the numerical implementation of the real-time forecasting model SLOSH (Jelesnianski and Chester, 1992) to improve
inundation performance in a coastal region with narrow channels. Woodruff et al. (2023) scaled up these approaches to the
entire South Atlantic Bight and showed improved performance of a subgrid model to a conventional high-resolution model for
Hurricane Matthew (2016).
While these advances have led to great improvements in estuarine and storm surge modeling, the assumption of hydraulic
connectivity of subgrid cells remains a challenge. To that end, Begmohammadi et al. (2021) removed the artifact of flows
occurring through catchment boundaries that are not resolved in a subgrid approach by restricting flow to a predetermined
path. Rong et al. (2023) introduced a new diffusive scheme in the existing subgrid channel approach to better model flood
routing in rivers and adjacent flood plains. Yu and Lane (2011) applied a subgrid approach to resolve the roughness effects of
small-scale structural elements in river floodplain cases, based on the method by Yu and Lane (2006) and applied a storage
correction to the coarser scale flow grid based on the higher-resolution topographic information accounting for cell blockage
and conveyance effects.
However, none of these efforts combined a reduced-complexity model with a subgrid approach that accounts for bed level and
friction variations for efficient compound flood modeling. In this paper, we explore a subgrid approach for the Linear Inertial
Equations (Bates et al., 2010) that are used in the SFINCS model (Leijnse et al., 2021). All model results were obtained with
the SFINCS 'Cauberg' release from November 2023 which is available as open-source code on GitHub and via
https://www.deltares.nl/en/software-and-data/products/sfincs (van Ormondt et al., 2023).  Computational speed is determined
by running the simulations on an Intel core I9 10980XE CPU.





The paper is organized as follows: we start with the governing equation in SFINCS, and a description of the new subgrid
approach (Section 2). We then demonstrate the accuracy of the subgrid method for some conceptual cases (Section 3). In
Section 4, the subgrid method is verified against the default SFINCS results and observed data for two real-world cases: tidal
propagation at the St. Johns River (Florida, USA) and the flooding during Hurricane Harvey (Houston, USA). The findings
are discussed in Section 5 and our conclusions are presented in Section 6.
**2    Model description**
**2.1    SFINCS governing equations**
The SFINCS model solves the shallow-water equations on a regular, staggered Arakawa-C grid. Its governing equations are
based on the Linear Inertial Equations (LIEs; Bates et al., 2010). In particular, the volumetric flow rate per unit width at the
interface between adjacent cells in the $x$ direction for the current time step is computed with Equation 1:
$$q_u^{t+\Delta t} = \frac{q_u^t - g\Delta t h_u \frac{\Delta z}{\Delta x} + F\Delta t}{1 + g\Delta t n^2 |q_u^t| / h_u^{7/3}}$$
(1)

where $q_u^t$ is the flow rate at the previous time step, $h_u$ and $\Delta z / \Delta x$ are the water depth and water level gradient at the cell interface
$u$, $g$ is the acceleration constant, $n$ is the Manning's n roughness and $\Delta t$ is the time step. The water depth $h_u$ at the cell interface
is computed in SFINCS as the difference between the maximum water level in the two adjacent cells and the maximum bed
level in these cells. For the sake of brevity, additional forcing terms, such as wind drag, barometric pressure gradients, and the
advection term, are represented in the combined term $F$.

The mass continuity equation reads:
$$z_{s\,m,n}^{t+\Delta t} = z_{s\,m,n}^t + \Delta t \left( \frac{q_{u\,m-1,n}^t - q_{u\,m,n}^t}{\Delta x} + \frac{q_{v\,m,n-1}^t - q_{v\,m,n}^t}{\Delta y} + \frac{S_{m,n}}{\Delta x \Delta y} \right)$$
(2)

where $z_s$ is the water level in a grid cell (with index $m$ in x-direction, $n$ in y-direction), and $S_{m,n}$ is an (optional) source term in
$m^3/s$ (e.g. to represent precipitation or a user-defined point source). In the remainder of this document, formulations will often
be presented in the $x$ direction, with the $y$ direction treated analogously (with cell interface $v$).

SFINCS uses a first-order explicit backward in time with a first-order central difference approximation of the spatial derivatives
(BTCS-scheme).



## 2.2 Subgrid corrections in the momentum equation

The goal of the subgrid approach is to compute flooding in a computationally efficient way using larger grids while retaining information of the higher-resolution elevation data. This is achieved by adjusting the conveyance depth $h_u$ and Manning's roughness $n$ in Equation 1 based on the local water level $z_u$ and the subgrid topography and roughness so that the unit discharge $q_u$ through a cell interface equals the average of the unit discharge of the subgrid pixels within the considered velocity point. An important assumption here is that the water level within the velocity point is constant, and therefore equal for all subgrid pixels. If the subgrid topography is known, and we assume that the water level $z_u$ is constant for all subgrid pixels in the velocity point, then representative values for $h_u$ and $n$ (as well as the wet fraction $\varphi$) can be computed as a function of $z_u$ and stored in look-up tables for each velocity point. During a simulation, these look-up tables are queried at each time step to provide representative values for $h_u$, $n$, and $\varphi$. This Section explains the theory behind the subgrid approach for the LIEs. The following sections describe the practical generation of the subgrid tables, and how these are queried during a SFINCS simulation.

Following the notation of Kennedy et al. (2019), for a quantity $Q$, hydrodynamic variables coarsened to the grid scale are defined as:

$$\langle Q \rangle_G = \frac{1}{A} \iint_{A_W} Q \, dA \tag{3}$$

where $A_W$ is the wet portion of the grid cell area A. This will be called the "grid average" and is denoted with subscript "G".

On the other hand, the "wet average" of $Q$, denoted with subscript "W" is:

$$\langle Q \rangle_W = \frac{1}{A_W} \iint_{A_W} Q \, dA \tag{4}$$

With the wet average area is defined as:

$$A_W = \varphi A \tag{5}$$

where $\varphi$ is the wet fraction of the cell area, then for hydrodynamic quantity Q:

$$\langle Q \rangle_G = \varphi \langle Q \rangle_W \tag{6}$$

The LIEs in their subgrid form using wet average quantities can be defined as:

$$\langle q_u \rangle_W^{t+\Delta t} = \frac{\langle q_u \rangle_W^t - g \, \Delta t \, \langle H_u \rangle_W \frac{\Delta z}{\Delta x} + F \Delta t}{1 + g \, \Delta t \, n_{u,W}^2 \, |\langle q_u \rangle_W^t| / \langle H_u \rangle_W^{7/3}} \tag{7}$$





where $\langle q_u \rangle_W$ and $\langle H_u \rangle_W$ are the wet average unit discharge and water depth, respectively, and $n_{u,W}$ is the Manning's n
coefficient adjusted for subgrid variations.

The expression for $n_{u,W}$ can be derived by considering Manning's equation for open channel flow :

$$\langle q_u \rangle_W = \sqrt{i}\, \frac{\langle H_u \rangle_W^{5/3}}{n_{u,W}} \tag{8}$$

where $i$ is the water level slope $\frac{\Delta z_S}{\Delta x}$. In case of a stationary current and in the absence of external forcing, the subgrid form of
the LIEs reverts to Equation 8. Consider now a velocity point with $N$ subgrid pixels, each with its own bed level $z_{b,k,}$ and
roughness $n_k$ (see Figure 1 and Figure 2). For a water level $z_u$, the water depth in each pixel is $h_k = \max(z_u - z_{b,k}, 0)$. The wet
average unit discharge of the subgrid pixels within the velocity point is:

$$\langle q_u \rangle_W = \frac{1}{\varphi_u N}\sqrt{i} \sum_{k=1}^{N} \frac{h_k^{5/3}}{n_k} \tag{9}$$

where $\varphi_u N$ is the number of wet pixels. Equation 9 can also be written as:

$$\langle q_u \rangle_W = \sqrt{i}\, \langle \frac{H_u^{5/3}}{n} \rangle_W \tag{10}$$


Substituting Equation 10 into Equation 8 yields the expression for $n_{u,W}$ (Equation 11):

$$n_{u,W} = \frac{\langle H_u \rangle_W^{5/3}}{\langle \frac{H_u^{5/3}}{n} \rangle_W} \tag{11}$$


The subgrid form of the LIEs (Equations 7 and 11) can alternatively be expressed with grid average quantities. The SFINCS
model uses these to solve the momentum balance, rather than the wet average quantities described above. Although somewhat
less intuitive, using grid average quantities has a few practical advantages that will be discussed in the next section. To write
the subgrid form of the LIEs using grid average quantities we simply substitute $\langle q_u \rangle_W$ with $\langle q_u \rangle_G / \varphi_u$ and $\langle H_u \rangle_W$ with
$\langle H_u \rangle_G / \varphi_u$ in Equation 7:

$$\langle q_u \rangle_G^{t+\Delta t} = \frac{\langle q_u \rangle_G^t - g\,\Delta t\, \langle H_u \rangle_G \frac{\Delta z}{\Delta x} + \varphi_u F \Delta t}{1 + g\,\Delta t\, n_u^2\, \left| \langle q_u \rangle_G^t \right| / \langle H_u \rangle_G^{7/3}} \tag{12}$$

where $n_u$ is $\varphi_u^{2/3} n_{u,W}$.

Using the same logic as for Equation 11, $n_u$ (hereafter called the representative roughness) can also be written as:





$$n_u = \frac{\langle H_u \rangle_G^{5/3}}{\left\langle \frac{H_u^{5/3}}{n} \right\rangle_G} \qquad (13)$$


For a known subgrid topography, and assuming a constant water level $z_u$ for all subgrid pixels in the velocity point, $\langle H_u \rangle_G$, $n_u$,
and $\varphi_u$ can be stored in look-up tables as a function of $z_u$. The generation of such tables is a pre-processing step that occurs
only once when the model is set up, and is not repeated in the computational loop. First, a subgrid is generated that has the
same orientation as the coarser hydrodynamic grid and a higher resolution. The level of refinement of the subgrid is an even
integer and is typically chosen such that the subgrid resolution roughly equals that of the digital elevation model (DEM). Next,
the subgrid model bathymetry is generated by interpolating a high-resolution DEM onto the subgrid. The roughness values are
determined at the subgrid scale as well, for example by converting data from land use maps to Manning's n values and
interpolating these onto the subgrid. An example of topography and roughness on the subgrid at a velocity point is provided
in Figure 1. Specifically, the high-resolution subgrid topography and roughness values around a single velocity point
demonstrate that information from both sides (A and B) of the water level grid cell is included in calculating the flux over the
cell face $q_{u\,m,n}$ between $z_{m,n}$ and $z_{m+1,n}$.

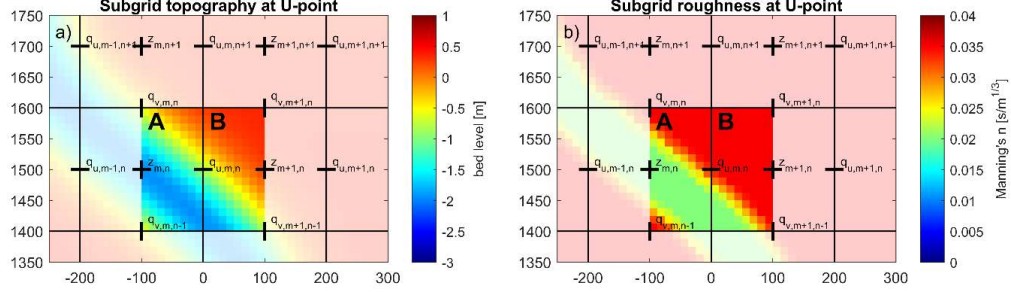


**Figure 1. High-resolution values of elevation $z$ (panel a) and roughness $n$ (panel b) at a U velocity point with a resolution of $N$=16×16**
**per computational cell. Colors for elevation and roughness indicate subgrid-scale values which are aggregated on the computational**
**black grid cells. Water level points are indicated by '+', while velocity points are marked with '–' and '|'.**

For each velocity point (here: u), we distinguish between two sides A and B of a computational cell (see Figure 1). The
minimum ($z_{b,A,min}$ and $z_{b,B,min}$) and maximum ($z_{b,A,max}$ and $z_{b,B,max}$) pixel elevations at both sides are determined. The combined
minimum and maximum elevations $z_{min}$ and $z_{max}$ are defined as:

$$z_{min} = max(z_{b,A,min}, z_{b,B,min}) \qquad (14)$$

$$z_{max} = max(z_{b,A,max}, z_{b,B,max}) \qquad (15)$$


Values of $\langle H_u \rangle_G$, $n_u$, and $\varphi_u$ are now computed at discrete equidistant vertical levels, ranging between $z_{min}$ and $z_{max}$ as



192    :

$$\varphi_{u,m} = \frac{1}{N} \sum_{k=1}^{N} p(z_m - z_{b,k}) \tag{16}$$

where $p(z_m - z_k)$ is 1 for $z_m > z_k$, and 0 for $z_m \leq z_k$:

$$\langle H_u \rangle_{G,m} = \frac{1}{N} \sum_{k=1}^{N} \max(z_m - z_{b,k}, 0) \tag{17}$$

$$n_{u,m} = \frac{\langle H_u \rangle_{G,m}^{5/3}}{\frac{1}{N} \sum_{k=1}^{N} \left( \max\left(z_m - \max(z_{b,k}, z_{min}), 0\right)/n_k \right)^{5/3}} \tag{18}$$

The number (*M*) of discrete vertical levels is defined by the user. We have found that around 20 levels are typically sufficient
to accurately describe the subgrid quantities $\langle H_u \rangle_G$, $n_u$ and $\varphi_u$ as a function of water levels between $z_{min}$ and $z_{max}$ and is used
throughout this paper. The vertical distance between each level is defined as $\Delta z = (z_{max} - z_{min}) / (M - 1)$, and the elevation of
each discrete level is $z_m = z_{min} + (m - 1) \Delta z$ (in which m goes from 1 to M).

The subgrid tables and resulting flux (panel d) for the velocity point depicted in Figure 1, using *M*=20 are illustrated in Figure
2. Red markers highlight the values at the discrete vertical levels.

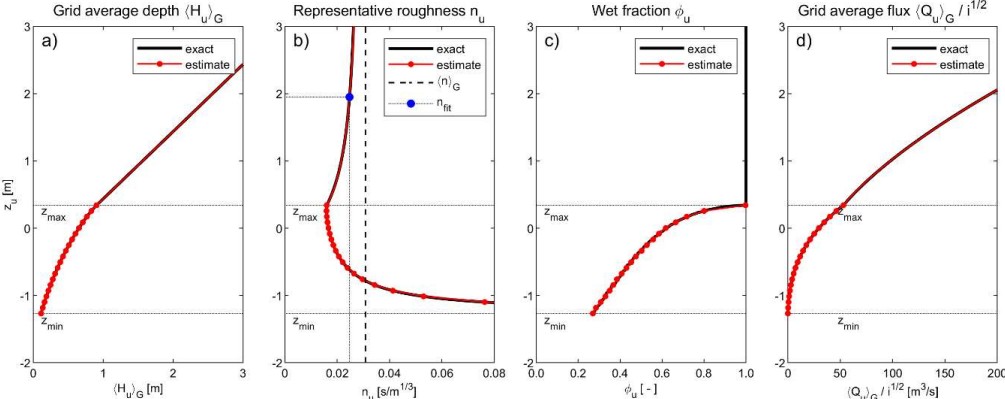


**Figure 2. Computation of subgrid quantities $\langle H_u \rangle_G$ (panel a), $n_u$ (panel b) and $\varphi_u$ (panel c) as a function of water level $z_u$ with 20
discrete vertical levels (M = 20). The resulting flux divided by the square root of the water slope I is shown in panel d. The black line
shows the exact solution obtained by solving Equations 5, 10, 11 and 17. The red line shows the estimate used in the SFINCS model,
with (for z <= $z_{max}$) linear interpolation of look-up table values, and (for z > $z_{max}$) linear increase for $\langle H_u \rangle_G$ and fit for $n_u$.**




Note that in Equation 18, to determine the representative roughness, the maximum of the pixel elevation and $z_{min}$ is used. This
is done to ensure that when the water level $z_u$ approaches $z_{min}$, i.e. when the highest of two adjacent grid cells becomes dry, $n_u$
will become very large, thereby effectively blocking flow between sides A and B. No water is allowed to flow when $z_u$ drops
below $z_{min}$.

The determination of $n_u$ for completely wet velocity points is more complicated, due to its non-linear relationship with $z_u$ at $z_u$
$> z_{max}$ (see Figure 2b). It would be possible to store values of $n_u$ at many levels above $z_{max}$ in the subgrid tables, but that could
result in too large file sizes and memory use. To avoid this, SFINCS uses the following estimation for $n_u$ instead:
$$n_u = \langle n \rangle_G - \frac{\langle n \rangle_G - n_{u,M}}{\beta(z_u - z_{max}) + 1}$$
(20)

where $\langle n \rangle_G$ is the average Manning's n of all subgrid pixels, and $\beta$ is a fitting coefficient (with both these parameters also
stored in the subgrid tables). The fitting coefficient $\beta$ is determined for each velocity point as:
$$\beta = \frac{\frac{\langle n \rangle_G - n_{u,M}}{\langle n \rangle_G - n_{fit}} - 1}{z_{fit} - z_{max}}$$
(21)


Here we have defined the level $z_{fit}$ at $z_{max} + (z_{max} - z_{min})$. The value for $n_{fit}$ at $z_{fit}$ is determined in a manner similar to Equation

224  18:

$$n_{fit} = \frac{\left(\langle H_u \rangle_{G,M} + z_{fit} - z_{max}\right)^{5/3}}{\frac{1}{N}\sum_{k=1}^{N}\left(\frac{z_{fit} - \max(z_{b,k}, z_{min})}{n_k}\right)^{5/3}}$$
(22)

The estimated value for $n_u$ above $z_{max}$ using Equation 20 is shown in Figure 2b, with the blue marker indicating $n_{fit}$. In very
deep water ($z_u >> z_{max}$), $n_u$ approaches $\langle n \rangle_G$, whereas for $z_u = z_{max}$, $n_u$ is equal to $n_{u,M}$.

The behavior of $n_u$ in Figure 2b can seem non-intuitive. Whereas the grid average water depth $\langle H_u \rangle_G$ has a real physical
meaning, the representative roughness nu should not be interpreted as a physical quantity but rather as a quantity that is used
to control the flux through a velocity point, given a certain grid average water depth and water slope i. It is a function not only
of the physical subgrid roughness but also of the subgrid water depth.

As mentioned previously, SFINCS uses grid average, rather than wet average quantities. Theoretically, both options would
yield identical results. The reason to choose a grid average approach is that the wet average depth and adjusted roughness can
vary much more rapidly and irregularly with changing water levels than their grid average equivalents. As a result, many more
vertical levels in the subgrid tables would be required to accurately describe wet average quantities as a function of z. This is





illustrated by considering a velocity point with a subgrid topography cross-section (Figure 3a). The average water depth and
adjusted roughness as a function of water level z (Figures 3a and 3b, respectively).

At each time step the model computes the water level zu at each velocity point using the maximum of the computed water
levels in the two adjacent cells, i.e. $z_u = \max(z_{s\,m,n}, z_{s\,m+1,n})$. This value is then used to query the look-up tables to find
appropriate values of the quantities $\langle H_u \rangle_G$, nu, and φu. For partially wet velocity points ($z_{min} < z_u < z_{max}$), a linear interpolation
of the values in the tables is used. When the entire velocity point is wet ($z_u \geq z_{max}$), the depth $\langle H_u \rangle_G$ increases linearly with zu:

$$\langle H_u \rangle_G = \langle H_u \rangle_{G,M} + z_u - z_{max} \tag{19}$$

**2.3    Subgrid corrections in the continuity equation**
The subgrid continuity equation is written in terms of grid average fluxes as:

$$V_{m,n}^{t+\Delta t} = V_{m,n}^t + \Delta t \left( \left( \langle q_u \rangle_{G,m-1,n}^t - \langle q_u \rangle_{G,m,n}^t \right) \Delta y + \left( \langle q_v \rangle_{G,m,n-1}^t - \langle q_v \rangle_{G,m,n}^t \right) \Delta x + S_{m,n} \right) \tag{23}$$

Contrary to Equation 2, Equation 23 computes the wet volume at the next time step, rather than the water level. The
corresponding water level $z_s$ is obtained from the continuity subgrid tables.

To generate the subgrid tables first the minimum and maximum pixel elevations $z_{min}$ and $z_{max}$, as well as the wet volume $V_{max}$
(defined as the wet volume between $z_{min}$ and $z_{max}$) are determined for each hydrodynamic grid cell (e.g. Figure 3). Then the
wet volume as a function of the local water level is determined:

$$V(z) = \frac{\Delta x \Delta y}{N} \sum_{k=1}^{N} \max(z - z_k, 0) \tag{24}$$

where N is the number of subgrid pixels in a grid cell. Finally, a number (*M*) of discrete equidistant volumes are defined,
ranging between 0 and $V_{max}$, where each volume is V$_m$ = *(m − 1)* $V_{max}$ / *(M - 1)*. By iterating over each discrete volume $V_m$, we
can (using linear interpolation of Equation 24) determine the corresponding water levels $z_s$. An example is given in Figure 3
which shows the volumes of the highlighted cell.



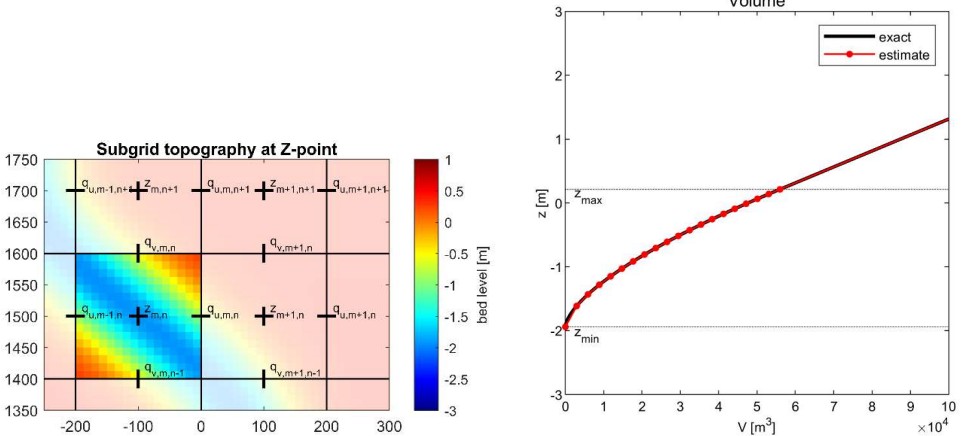

**Figure 3. Panel A: values on the subgrid-scale of elevation z at a water level point (N=16x16). Panel B. Representation of water level $z_s$ as a function of volume V with 20 discrete volumes (M = 20). The black line shows the exact solution of Equation 24. The red line shows the estimate of $z_s$ used in the SFINCS model with, for $z_s <= z_{max}$, linear interpolation of look-up table values, for $z_s > z_{max}$ a linear increase with V.**

During a simulation, the model computes at each time step the volume $V$ in each cell and queries the look-up tables to find the matching value for $z_s$. For partially wet cells ($V < V_{max}$), a linear interpolation of the values in the tables is used. When the entire cell is wet ($V \geq V_{max}$), the water level $z_s$ increases linearly with $V$ and is computed as

$$z_s = z_{max} + \frac{V - V_{max}}{\Delta x \Delta y} \tag{25}$$

Note that for pre-processing purposes, it would have been more straightforward to describe the wet volume $V$ at equidistant vertical levels $z_m$ (similar to the approach for the momentum subgrid tables). However, during the simulation, the linear interpolation of subgrid data with equidistant volume levels is much more efficient.

## 2.4    Pre and post-processing

Pre-processing steps for SFINCS include creating a mask file describing (in)active cells, interpolating bathymetry and roughness values, and imposing boundary conditions. Tools to carry out these steps are available in both Delft Dashboard (Van Ormondt et al., 2020) and HydroMT-SFINCS (Eilander et al., 2023 or https://deltares.github.io/hydromt_sfincs/latest/), which both also have the capability to generate subgrid table files using high-resolution DEMs.

SFINCS stores the output of hydrodynamic quantities on the (coarse) computational grid. These results can be further downscaled to higher-resolution flood maps at the original DEM resolution (assuming again that the computed water level in





a grid cell is representative of each subgrid pixel within that cell). Flood depths at the DEM scale are computed by subtracting
the elevation of each DEM pixel from the water level in the cell. An example of the results is presented in Figure 10.

## 3   Conceptual verification cases: straight and meandering channels

The first conceptual test involves a 5 km long straight channel of 100 m wide with 1:5 side slopes (Figure 4a and c), for which
a synthetic bathymetry was created. The slope of the channel is $10^{-4}$ downhill in y-direction, and the flood plains on either side
of the channel have an elevation of 0.3 m above the water level in the channel. The Manning's n roughness is set to 0.02 s/m$^{1/3}$.
Water level boundary conditions at the upstream and downstream sides are set to +0.25 m and -0.25 m, respectively, resulting
in a $10^{-4}$ water level slope, equal to the channel slope. The analytical solution, using Manning's equation for open channel flow
yields a discharge of 360 m$^3$/s. The input files for the 5m subgrid version of this model setup can be found in Appendix B1.

The second test is identical to the first, except that it has a meandering channel. The meandering channel has a sinuosity $\Omega$ of
1.32, i.e. the ratio between the length along the channel (6603 m) and its straight-line length (5000 m) (see e.g. Lazarus and
Constantine, 2013 for background on river sinuosity). As the water levels upstream and downstream of the channel are kept
the same, the water level slope in the meandering channel is smaller by a factor $\Omega$, resulting in a (lower) analytical discharge
of 313 m$^3$/s.

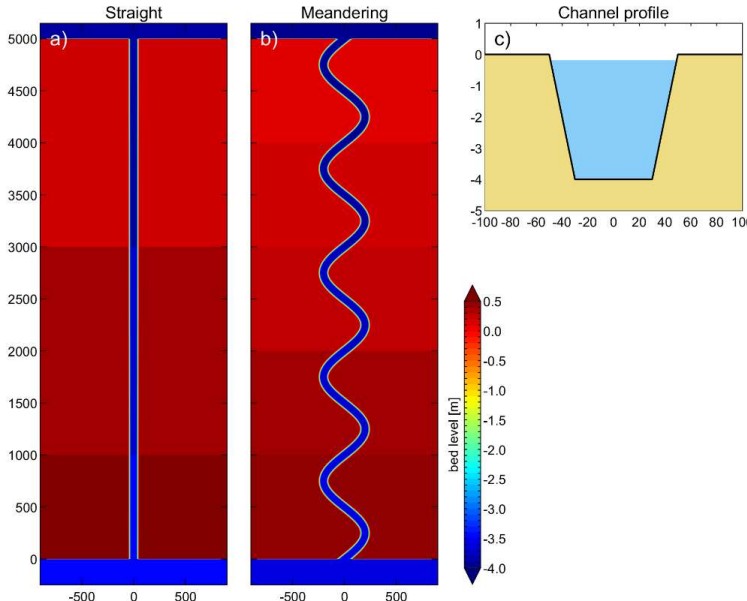


**Figure 4. Schematized channel used in the conceptual verification cases, including a straight channel (top view, panel a), a meandering channel (top view, panel b), and a cross-section (panel c).**

Simulations are carried out at various grid resolutions (5, 10, 20, 50, 100, 200, and 500 m), with both the subgrid method and regular versions of SFINCS. The subgrid simulations use a 1 m resolution subgrid, onto which the DEM is bilinearly interpolated. For the regular topography simulations, grid cell averaging is used to schematize the model bathymetry, in which the bed level of each cell is set equal to the mean of the DEM pixels within that cell. Figure 5 shows the regular model bathymetry at grid resolutions Δx of 10 m, 50 m, and 200 m for the meandering channel. It is clear that whereas the first two capture the channel topography reasonably well, the channel depth in the 200 m model is strongly underestimated, and its width is proportionally overestimated.





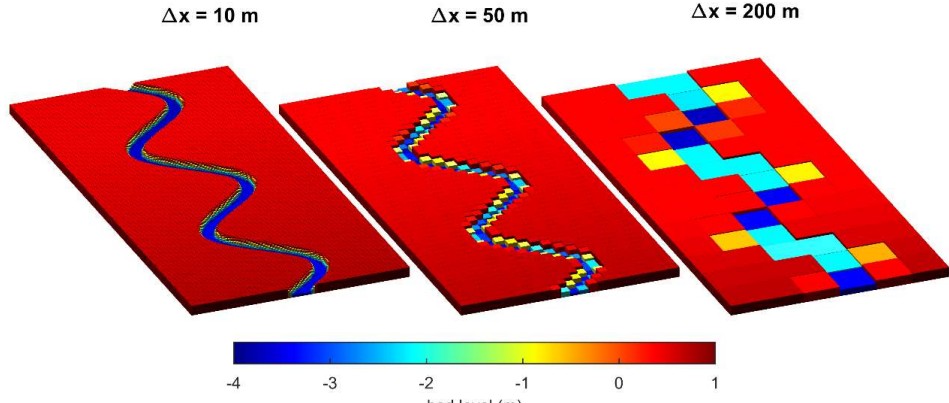

**Figure 5 Schematized meandering channel bathymetry with regular topography for hydraulic grid resolutions Δx = 10 m, Δx = 50 m, and Δx = 200 m**

In the first test (straight channel), the regular bathymetry models stay reasonably close to the analytical solution up to resolutions of 50m (blue bars in Figure 6 – panel A). The accuracy of the coarser models however degrades significantly with decreasing grid resolution as is to be expected. The channel depth in the coarser models is increasingly underestimated, and even though its width is proportionately overestimated, the strongly non-linear relationship between water depth and discharge results in a decrease of the discharge with decreasing grid resolution. In contrast, the discharges computed by the subgrid models are within 2% of the analytical solution across all grid resolutions (red bars in Figure 6 – panel A), proving that, at least for very simple conceptual cases, the subgrid method presented here is accurate.



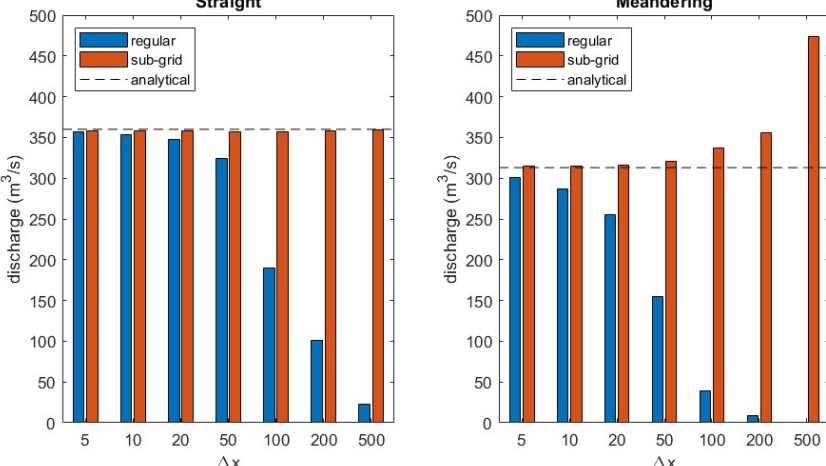

315

**Figure 6. Effect of grid resolution Δx on computed discharges for regular and subgrid topography in straight (panel a) and meandering (panel b) channel.**

In the second test (meandering channel), the trend of the regular models is similar to those in the first test (blue bars in Figure 6 – panel B), but the performance is lower than in the straight channel case, with the discharge for the two coarsest regular models going to zero. This is caused by the fact that the hydraulic connection between some channel cells is broken in the coarsest models (see also Figure 5).

The subgrid models in the second test show very good accuracy at resolutions up to 50 m. Coarser models start to overestimate the discharge. The 500 m model in particular computes a discharge of 473 m$^3$/s (an overestimation of the analytical discharge by ~51%). There are two reasons for this: as the coarse mesh does not capture the scale of the meanders, the channel is effectively schematized as a straight channel with a length of 5000 m. This leads to an overestimation of the true water level slope and resulting wet average flux. Secondly, meanders inside a grid cell result in a larger wet fraction, which the model "interprets" as a wide channel, leading to a further overestimation.

For rivers with meanders that are not resolved by the model grid, we can approximate the discharge overestimation as a function of the channel sinuosity:

$$\frac{Q_m}{Q_r} = \Omega^{3/2} \tag{26}$$

where $\Omega$ is the sinuosity, $Q_r$ is the true discharge and $Q_m$ is the discharge computed with the subgrid method (see Appendix A for the derivation of Equation 26). Equation 26 suggests that the discharge overestimation in the 500 m subgrid model (which



does not resolve the meandering at all) is ~52 % ($1.32^{3/2}$), which closely matches the computed overestimation of ~51%
reported earlier.
**4    Real-world application cases**
**4.1    Tidal propagation St. Johns River**
Leijnse et al. (2021) described SFINCS model results for Hurricane Irma (2017) along the St. Johns River (Florida, USA). The
length of the river is about 170 kilometers from its mouth to Lake George upstream (Figure 7 – panel A) where still a small
tidal signal remains. Its width varies between 400 m and 5 km. Although the model showed good skill when compared to a
full-physics Delft3D model, its 100-meter grid resolution proved insufficient to adequately propagate the tide into the estuary.

In this test case, the St. Johns River SFINCS model from Leijnse et al. (2021) is adapted and tidal propagation into the river is
simulated at several horizontal resolutions (25, 50, 100, 200, and 500 m) using both the regular and subgrid approach. The
topography and bathymetry data are improved by using data obtained from the Continuously Updated Digital Elevation Model
(CUDEM; CIRES, 2014). The Manning friction coefficient in the river is set to 0.02 s/m$^{1/3}$. The offshore boundary water levels
are derived from TPXO 8.0 tidal components (Egbert and Erofeeva, 2002). Computed water levels are validated against
observed tidal components from 11 tide stations (retrieved through Delft Dashboard; van Ormondt et al., 2020) (Figure 7 –
panel A). The input files for the 25m subgrid version of this model setup can be found in Appendix B2.

Simulations are carried out over a one-month period to assess the model's capability to propagate the tide into the river.
Analysis of the main tidal component M2 across different model variations reveals considerable differences in the upstream
propagation (Figure 7B). The amplitude of M2 is approximately 75 cm at the offshore boundary and sharply decreases near
the city of Jacksonville, where the river narrows significantly (about 40 kilometers upstream along the river). At 100-meter
resolution, the SFINCS model with regular topography can reproduce the main trends but underestimates the tidal amplitudes
relative to observations (Figure 7B), as in Leijnse et al. (2021). At the coarser 500-meter resolution, this underestimation of
amplitude is significantly stronger and the tide arrives too late (Figure 7C).  The tidal propagation only accurately matches the
observations when utilizing a 25-meter resolution with the regular topography.

**The subgrid version of SFINCS, on the same 100-meter grid resolution, mitigates the underestimation of the regular (non-subgrid)**
**version (Figure 7B). The median error of M2 amplitude prediction over the 11 observation stations decreases from 2.6 cm to 0.4 cm,**
**the phase error from 4.1 to 2.1 degrees, and the overall RMSE from 8.0 to 6.4 cm (Overview of the St. Johns River near Jacksonville,**
**FL, USA (Panel A), with analysis points (green dots) and tide gauges (yellow dots). Panel B: Observed (black dots) and modeled**
**(colors) M2 tidal amplitudes along the river from downstream to upstream. Panel C: Observed (black dots) and modeled (colors)**
**M2 tidal phases along the river. Different colors represent variations in the SFINCS model setup: red indicates the regular non-**
**subgrid version, while blue denotes the subgrid version, with decreasing color intensity indicating a decrease in model resolution.**
**M2 phase is converted from degrees to hours, assuming one degree equals 12.42 hours / 360 degrees. The coordinate system is WGS**
**84 / UTM 15 N (EPSG 32615).**



Table 1). Further analysis of different grid resolutions via the subgrid method illustrates that, even with coarser grid resolutions,
the subgrid-enabled SFINCS version propagates the tide inland properly, even at very coarse resolutions of 500 meters. The
tidal phasing is also generally more accurately resolved with subgrid versus the regular SFINCS mode.

**Computing the RMSE over the whole month tidal prediction shows that error increases from about 8 cm to about 20 cm for coarser**
**grid resolutions in regular SFINCS mode (Overview of the St. Johns River near Jacksonville, FL, USA (Panel A), with analysis**
**points (green dots) and tide gauges (yellow dots). Panel B: Observed (black dots) and modeled (colors) M2 tidal amplitudes along**
**the river from downstream to upstream. Panel C: Observed (black dots) and modeled (colors) M2 tidal phases along the river.**
**Different colors represent variations in the SFINCS model setup: red indicates the regular non-subgrid version, while blue denotes**
**the subgrid version, with decreasing color intensity indicating a decrease in model resolution. M2 phase is converted from degrees**
**to hours, assuming one degree equals 12.42 hours / 360 degrees. The coordinate system is WGS 84 / UTM 15 N (EPSG 32615).**





Table 1). However, when incorporating subgrid corrections this remains stable around this value of 8 cm. While high tide peak
predictions remain robust for the subgrid SFINCS version at larger grid resolutions (Table 1), the performance decreases more
significantly for low water peaks, indicating that during these periods, the low tide flushing of the river is still underestimated.
Integrating the subgrid raises computational costs by around 0-72% (44% on average) as a result of the extra overhead involved
in querying the subgrid tables.

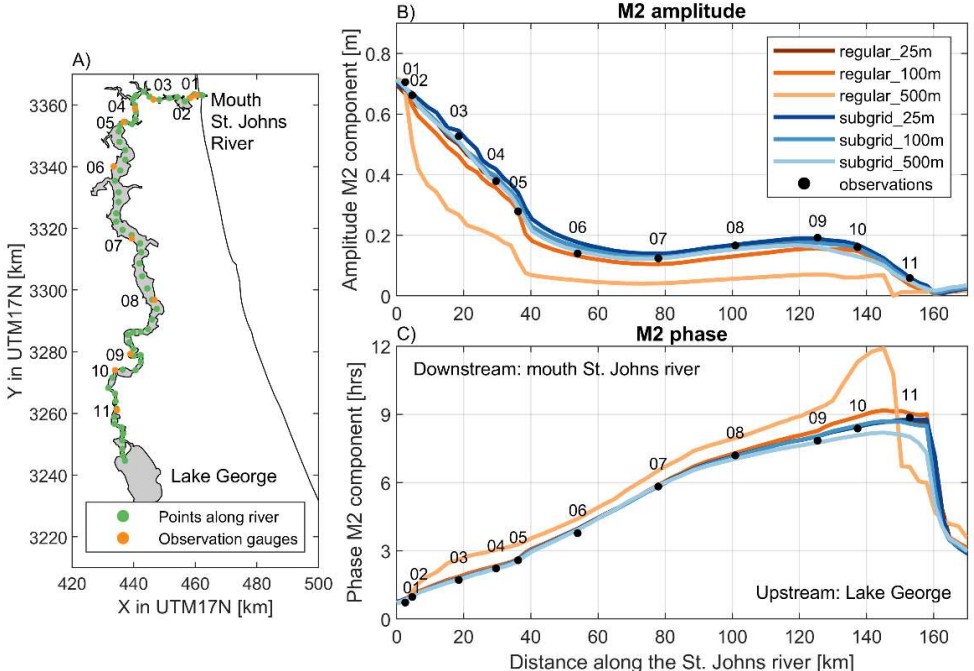


**Figure 7. Overview of the St. Johns River near Jacksonville, FL, USA (Panel A), with analysis points (green dots) and tide gauges**
**(yellow dots). Panel B: Observed (black dots) and modeled (colors) M2 tidal amplitudes along the river from downstream to**
**upstream. Panel C: Observed (black dots) and modeled (colors) M2 tidal phases along the river. Different colors represent variations**
**in the SFINCS model setup: red indicates the regular non-subgrid version, while blue denotes the subgrid version, with decreasing**
**color intensity indicating a decrease in model resolution. M2 phase is converted from degrees to hours, assuming one degree equals**
**12.42 hours / 360 degrees. The coordinate system is WGS 84 / UTM 15 N (EPSG 32615).**



**Table 1. Overview of model skill and computational expense for evaluated scenarios of inland tidal propagation at the St. Johns River, FL. Metrics include RMSE of overall difference in time-series compared to observations, RMSE of high water peaks, RMSE of low water peaks, difference in M2 amplitude, and difference in M2 phase, all presented as medians over 11 observation stations. The last column shows the runtime in seconds, measured on an Intel Core i9-10980XE CPU.**

| Run | RMSE overall [cm] | RMSE high water peak [cm] | RMSE low water peak [cm] | Amplitude difference M2 [cm] | Phase difference M2 [°] | Model runtime [sec] |
|---|---|---|---|---|---|---|
| regular_25m | 7.7 | 6.6 | 9.1 | -0.3 | 1.0 | 64512 |
| regular_50m | 7.8 | 5.7 | 10.1 | -1.7 | 5.0 | 7596 |
| regular_100m | 8.0 | 4.3 | 12.5 | -2.6 | 4.1 | 727 |
| regular_200m | 12.0 | 5.3 | 19.5 | -6.7 | 6.5 | 110 |
| regular_500m | 16.1 | 8.3 | 25.4 | -10.9 | 21.4 | 28 |
| regular_1000m | 20.1 | 14.5 | - | -15.9 | 50.1 | 11 |
| subgrid_25m | 8.7 | 8.3 | 7.3 | 1.5 | 1.2 | 98806 |
| subgrid_50m | 7.5 | 7.6 | 6.1 | 0.6 | 1.5 | 12127 |
| subgrid_100m | 6.4 | 5.3 | 6.1 | -0.4 | 2.1 | 1251 |
| subgrid_200m | 7.8 | 7.3 | 8.2 | -1.0 | 1.5 | 159 |
| subgrid_500m | 8.2 | 6.6 | 8.7 | -0.3 | -1.5 | 28 |
| subgrid_1000m | 7.8 | 7.1 | 8.5 | 0.7 | -4.7 | 15 |





## 4.2    Pluvial flooding during Hurricane Harvey


Sebastian et al. (2021) used SFINCS to hindcast the flood extent and flood depth during Hurricane Harvey (2017) in Houston,
TX. The model was validated against water level time series at 21 United States Geological Survey (USGS) observation points
and 115 high water mark (HWM) locations (Figure 8). The original model was run with a regular 25-meter resolution grid
based on a high-resolution continuous topo-bathymetry across the area of interest. The model had a fair correlation with
observed time series and HWM across the study area.

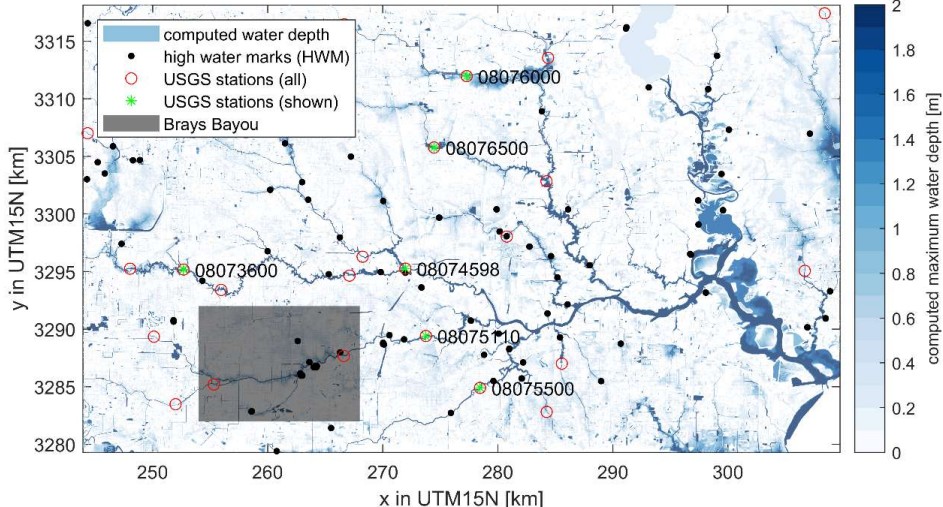


**Figure 8. Modeled flood inundation in the urban areas of Houston, TX, simulated with SFINCS at a 25m resolution with subgrid**
**corrections. Water depths less than 0.10 m are excluded for clarity. USGS stream gauges (red) and high-water marks (HWMs, black)**
**used for model validation are shown as solid circles. Six USGS stations, presented as time series in Figure 9, are marked with circles**
**and stars, including their station numbers. A zoom-in of the midstream portion of Brays Bayou is shown in Figure 10. The coordinate**
**system is WGS 84 / UTM 15 N (EPSG 32615). © Microsoft.**

In this field case, the model setup is adapted and flooding across Houston is simulated at several horizontal resolutions. In
particular, three variations for regular SFINCS (25, 50, and 100 meters) and 5 variations of subgrid (same resolutions as regular
mode, including 200, and 500 meters) were created. Model settings were based on Sebastian et al. (2021) model except for the
model resolution. Friction and infiltration capacity were cell-averaged from the original setup for the coarser model runs. The
input files for the 25m subgrid version of this model setup can be found in Appendix B3.





Almost all model versions reproduce the general shape of the observed hydrograph. However, the coarser regular version of
SFINCS results in larger errors mainly due to an overestimation of the water level (Figure 9). The overestimation is driven by
an incorrect representation of the bed level which is averaged across larger areas and can therefore not depict the local bayous
with coarser grid cells. SFINCS with the subgrid corrections improves the model skill (Table 2). For example, when comparing
the 25-meter regular with the subgrid-enabled SFINCS model with the same computational resolution, the Nash-Sutcliffe
Efficiency(NSE) increases from 0.35 to 0.58. NSE is a statistical metric used to evaluate the predictive accuracy of models by
comparing observed and predicted values. NSE values range from 0 to 1, with values closer to 1 indicating a better-performing
model. An NSE value of 0 means the model's predictions are as accurate as using the mean of the observed data as the predictor.
Model skill increases because more topo-bathymetry information is considered per grid cell via the subgrid correction in the
momentum and continuity equations (see Sections 2.2 and 2.3). Despite the subgrid correction, model skill still decreases with
decreasing computational resolution. For example, a 500-meter simulation with subgrid correction has an NSE close to zero.
Including the subgrid feature increases computational expense by 73 to 184 % (average of 129%), because of additional
overhead in querying the subgrid tables. The highest model skill is obtained with the finest model resolution (25m used here)
including subgrid. Selecting the model resolution of choice is a balancing act between model skill and computational expense.

SFINCS can store the maximum computed water level across the computational domain, with the capability to downscale this
data to higher-resolution flood maps as part of a post-processing step. In particular, to calculate flood depths at the DEM scale,
the elevation of individual DEM pixels is subtracted from the corresponding cell's water level (see Section 2.4). For instance,
the results demonstrate that the 25-meter resolution outcomes and those downscaled to a 100-meter subgrid are quite similar.
This is illustrated in Figure 10, which shows modeled flood inundation in the midstream portion of Brays Bayou using four
different SFINCS model options. Panels A and C in Figure 10 highlight the comparison: Panel A presents the regular 25-meter
resolution, while Panel C depicts the 'subgrid 100m – downscaled' method, which applies a downscaling method to the DEM
resolution as a post-processing step. However, the 100-meter subgrid resolution runs 35 times faster than the 25-meter regular
SFINCS version, while maintaining a similar level of accuracy (see Table 2) and thus, producing comparable extents of
flooding. Nonetheless, it is important to note that the 100-meter resolution results tend to provide a coarser visual representation
of flood extents, often overestimating them (see panels B and D in Figure A1) for both the regular and subgrid versions of
SFINCS.





**Table 2. Overview of model skill and computational expense for evaluated scenarios of pluvial flooding during Harvey. Model skill**
**metrics for time series, including NSE (Nash-Sutcliffe Efficiency), MAE (Mean Absolute Error), RMSE (Root Mean Square Error),**
**and bias, as well as MAE for high-water marks (HWMs). The last column shows the runtime in seconds, measured on an Intel Core**
**i9-10980XE CPU.**

| | Time series | | | | HWM | |
|---|---|---|---|---|---|---|
| simulation | NSE [-] | MAE [m] | RMSE [m] | bias [m] | MAE [m] | run time [s] |
| regular_25m | 0.349 | 1.68 | 2.14 | -0.548 | 0.73 | 12136 |
| regular_50m | -0.007 | 2.08 | 2.58 | 0.405 | 0.68 | 3552 |
| regular_100m | -1.988 | 3.41 | 3.94 | 2.493 | 0.84 | 116 |
| subgrid_25m | 0.581 | 1.29 | 1.58 | -0.842 | 0.89 | 20951 |
| subgrid_50m | 0.540 | 1.3 | 1.57 | -0.963 | 0.94 | 2801 |
| subgrid_100m | 0.495 | 1.35 | 1.62 | -0.984 | 0.98 | 341 |
| subgrid_200m | 0.310 | 1.62 | 1.94 | -1.226 | 1.09 | 38 |
| subgrid_500m | 0.011 | 2.05 | 2.47 | -1.671 | 1.27 | 6 |



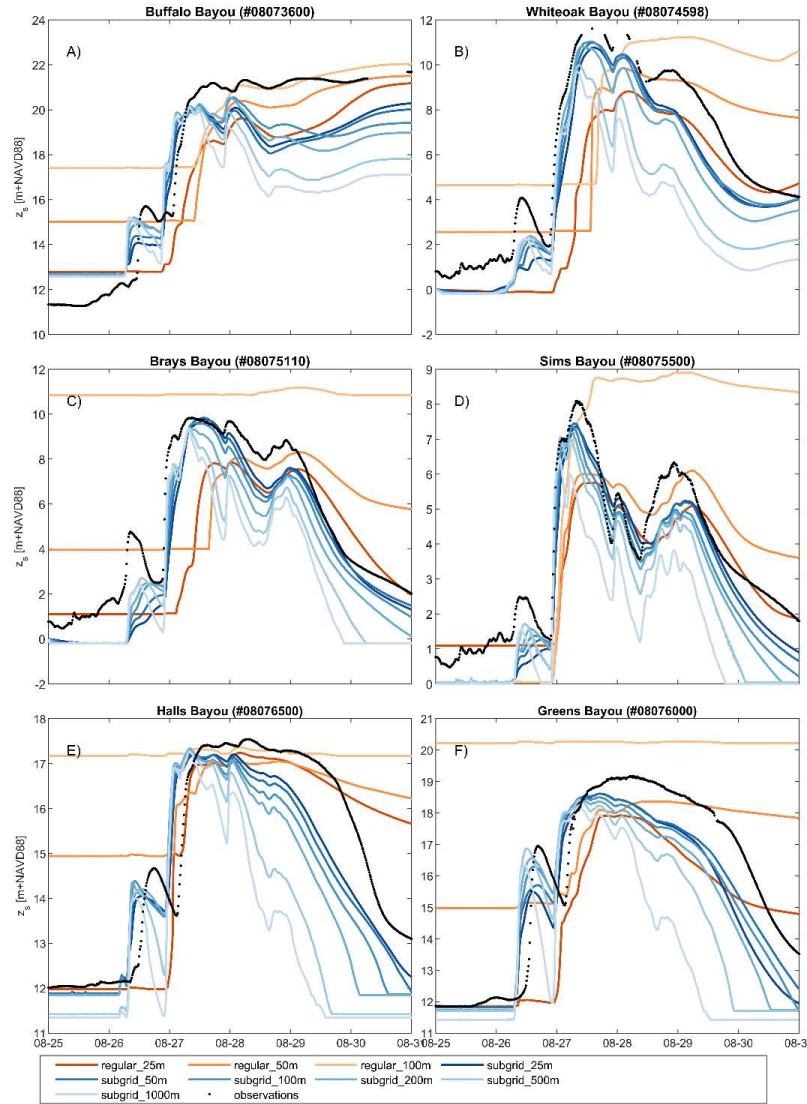

**Figure 9. Overview of (computed) water levels during Hurricane Harvey. Comparison between modeled (colored lines) and observed (black lines) hydrographs at six USGS gauge locations (labeled in Figure 8): Panels A. Buffalo Bayou (USGS 08073600); B. White Oak Bayou at Main Street (USGS 08074598); C. Brays Bayou at MLK Jr. Blvd (USGS 08075110); D. Sims Bayou at Houston, TX (USGS 08075500); E. Vince Bayou at Pasedena, TX (USGS 08075730); f Greens Bayou nr Houston, TX (USGS 08076000). Different colors represent variations in the SFINCS model setup. Red is used for the regular version of SFINCS (non-subgrid). Blue is used for the subgrid version of SFINCS. Decreasing color intensity depicts a decrease in model resolution.**



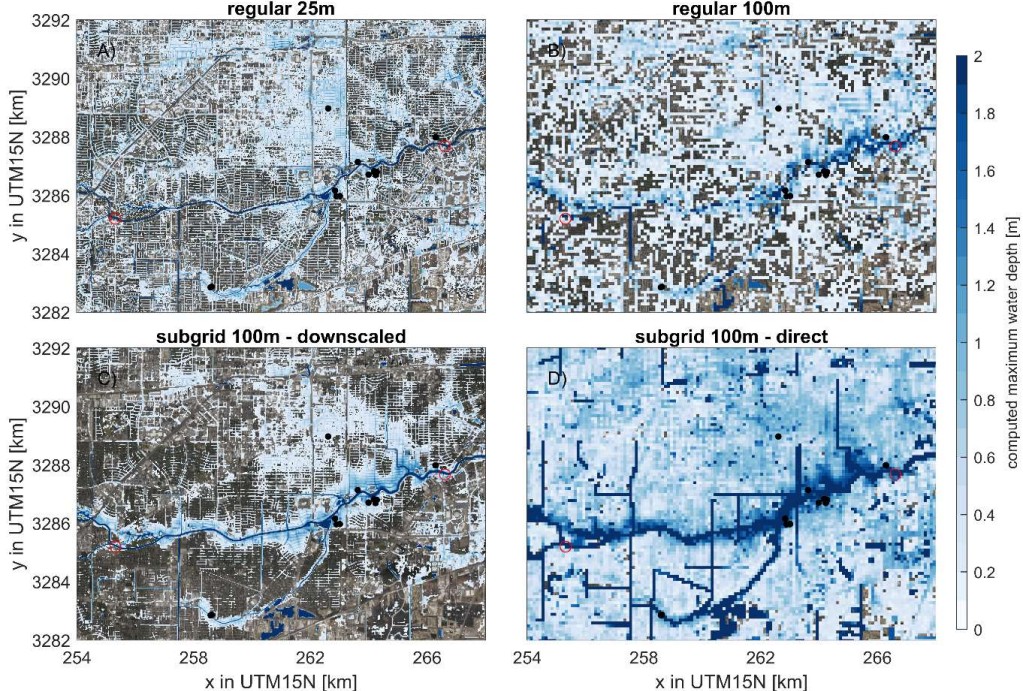

**Figure 10. Modeled flood inundation in the midstream portion of Brays Bayou for 4 different SFINCS model options: A) regular 25m, b) regular 100m, c) 'subgrid 100m – downscaled' is using the same model simulation as 'subgrid 100m – direct' (panel D), but then applying a downscaling method to the DEM resolution as a post-processing step. Water depths less than 0.10 m have been excluded for visual purposes. The locations of USGS stream gauges (red) and HWMs (black) used for the model validation are shown as solid circles.  The coordinate system of this figure is WGS 84 / UTM 15 N (EPSG 32615). © Microsoft.**





## 5   Discussion

The integration of subgrid corrections into SFINCS has led to significant enhancements in accuracy, as evidenced in both conceptual verification cases (Section 3) and real-world scenarios, including tidal propagation (Section 4.1) and pluvial flooding (Section 4.2). This section delves into the impact of these accuracy enhancements and outlines the remaining challenges and areas for future research, particularly concerning flow-blocking features and the overestimation of fluxes in meandering systems.

The ability to achieve improved accuracy on the same grid resolution signifies progress. However, in practical terms, a more accurate simulation also allows for the use of coarser model resolutions. This is particularly advantageous given SFINCS's explicit numerical scheme, enabling faster and thus more efficient compound flood modeling. For example, in the real-world application cases of tidal propagation (Section 4.1) and pluvial flooding (Section 4.2), a subgrid model at 100-meter resolution demonstrates comparable, if not higher, performance to the regular 25-meter resolution SFINCS model. However, the computational cost is significantly lower with a factor of 35 to 50 speedup. The introduction of subgrid corrections does introduce additional computational expenses versus regular SFINCS. For identical model resolutions, the inclusion of subgrid corrections for momentum and continuity results in an increase in computational costs by 44 to 129%.

The downscaling routines implemented also allowed for the use of the high-resolution data in the post-processing step. However, the simple subtraction of the computed water level and high-resolution topography (introduced in Section 2.4 and applied in Section 4.2) might result in water in an area that would not be flooded using high-resolution models. While this might not affect the accuracy compared to water level stations, it does influence results and flood extents. In particular, disconnected grid cells might pop up behind levees and other flow-breaking features which form a challenge when communicating the results to stakeholders.  Moreover, the presented downscaling routine has limited use for areas with steep gradients where the assumption of a constant water level per computational cell is invalid. Therefore, exploring more sophisticated hybrid surrogate models might improve the dynamic evolution of the flood extent (Fraehr et al., 2022).

Addressing subgrid connectivity poses a significant challenge for the implementation described in this paper and the broader modeling community. In contrast to approaches that relied on cell and edge clones (Begmohammadi et al., 2021) or artificial diffusion (Rong et al., 2023), SFINCS employs a subgrid weir formulation. This formulation, which is applied snapped to the grid, controls the flow between two cells but requires the creation of subgrid features during a pre-processing phase. To date, these features have been manually identified. However, there is ongoing research into algorithms capable of detecting flow-blocking features as well as the integration of methods from existing literature or direct modifications to the subgrid lookup tables to account for this.





Similarly, the overestimation of fluxes in situations with unresolved meanders continues to be a challenge. This issue is not
exclusive to SFINCS's implementation of subgrid corrections but is a common challenge across subgrid modeling. Various
estimates for the sinuosity $\Omega$ have been reported in scientific literature. Lazarus and Constantine (2013) suggest that the typical
range for $\Omega$ lies between 1 and 3, where 1 corresponds to a straight channel and 3 represents the upper limit for natural, freely
migrating meandering rivers. Hence, when using a computational grid that does not resolve the river meanders, the presented
subgrid approach may overestimate discharges by more than a factor of 5 (or $3^{3/2}$). To avoid this, it is recommended that the
grid spacing of the computational grid does not exceed the width of the river channel.
**6     Conclusions**
Large-scale flood models require high accuracy at acceptable computational times. One strategy to achieve this is to use
information available at a higher resolution than the hydrodynamic grid resolution in models through subgrid corrections. This
paper describes a set of subgrid corrections to the Linear Inertial Equations (LIE) using grid average quantities (depth,
representative roughness, wet fraction, and flux to the momentum equations and for the wet volume in the continuity equation)
which were implemented in SFINCS. The model uses pre-processed subgrid tables that correlate water levels with
hydrodynamic quantities by assuming constant water levels for all subgrid pixels.

The conceptual case of a straight channel showed good skill in terms of discharge fluxes with the subgrid model regardless of
the model resolution while the accuracy of the regular models without subgrid correction decreased significantly with
decreasing resolution. For the meandering channel differences start to emerge for coarser model resolutions with and without
subgrid corrections. In particular, the difference in discharge estimation was overestimated by 50% for the coarsest subgrid
model used.  The ratio between the length along the channel and its straight-line length (also known as sinuosity or $\Omega$) served
as a valuable metric for quantifying flux overestimations. The conceptual cases gave confidence that the corrections were
correctly implemented while also highlighting their limitations in grids that do not adequately resolve river meanders. In
particular, we introduced an equation that allows for approximation of the discharge overestimation as a function of the channel
sinuosity:

Real-world application cases further validated the subgrid corrections' benefits. For tidal propagation in the St. Johns River,
the subgrid model with a 500-meter resolution matched the accuracy of the 25-meter standard SFINCS model. Similarly, in
modeling pluvial flooding during Hurricane Harvey, a 25-meter resolution SFINCS model was necessary to achieve a Nash–
Sutcliffe Efficiency (NSE) of 0.35, while the subgrid variant at the same resolution outperformed this with an NSE of 0.58
(where a score of 1 would be perfect) and maintained comparable accuracy even at a coarser 100-meter resolution. Overall,
the implementation of subgrid corrections for LIE within SFINCS shows promise for enhancing model accuracy and reducing



computational demands in compound flooding simulations, marking a significant step forward in the field of hydrodynamic
modeling.

*Code and data availability.*
The SFINCS code is freely available to anyone and published on Zenedo (https://zenodo.org/doi/10.5281/zenodo.8038533)
and GitHub (https://github.com/Deltares/SFINCS).

*Author contributions.*
MO is the primary developer of the SFINCS model. KN, RG, and TL have actively contributed to the development of the
model. AvD initiated and co-wrote this paper. All authors were actively involved in the interpretation of the model outcomes
and the writing process.

*Competing interests.*
The authors declare that they have no conflict of interest.

*Acknowledgments and financial support*
We acknowledge the Deltares research program "Natural Hazards" which has provided funding to develop the model and write
this paper.



**Appendices**
**Appendix A: Derivation of discharge overestimation due to unresolved meandering**
The subgrid approach presented in this paper may result in an overestimation of fluxes between grid cells in places where river
meanders are not sufficiently resolved by the computational grid. The overestimation may be expressed as the ratio between
the computed and theoretical fluxes. In this appendix, we describe a simple relation between this ratio and the river sinuosity
in cases where the model grid does not resolve the meanders at all. The sinuosity is defined as the ratio between the length
along the channel and its straight-line length (e.g. Lazarus and Constantine, 2013).

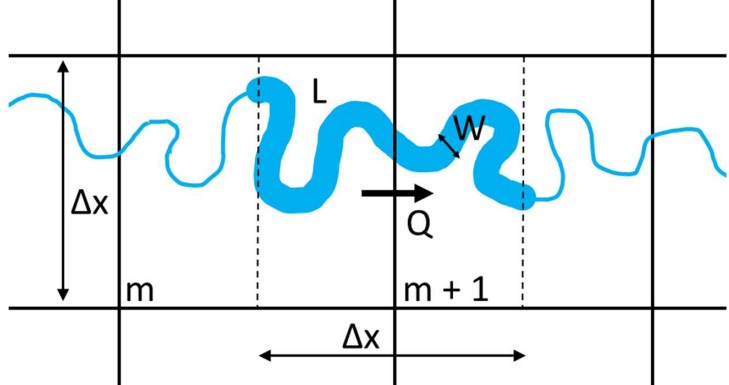

**Figure A1. Conceptual figure of the sinuosity which is a defined as the ratio between the length along the channel and its straight-**
**line length**
Using Manning's formula, the theoretical discharge can be described with:
$$Q_r = \frac{W\sqrt{\frac{\Delta z}{L}}H^{5/3}}{n}$$
$(A.1)$

where W is the river width, L is the length of the center line of river stretch, $\Delta z$ is the water level difference over the river
stretch, H is the channel depth (assumed uniform), and n is the Manning's roughness coefficient.
Inside a model using the subgrid method, the discharge computed at the cell interface will be:
$$Q_m = \Delta x \frac{\varphi\sqrt{\frac{\Delta z}{\Delta x}}H^{5/3}}{n}$$
$(A.2)$

where $\Delta x$ is the grid size, $\varphi$ is the wet fraction of the velocity point, and H is the "wet-average" depth.
We assume here that the sinuosity is:
$$\Omega = \frac{L}{\Delta x}$$
$(A.3)$

Furthermore, the wet fraction $\varphi$ in A.2 can be written defined as the river area W x L divided by the cell area:



$$\varphi = \frac{WL}{\Delta x^2} = \frac{W}{\Delta x}\Omega \qquad (A.4)$$

After substituting φ in Eq. A.2 with Eq. A.4, we can write the overestimation (i.e. the ratio of the computed and theoretical
discharge $Q_m$ / $Q_r$) as:
$$\frac{Q_m}{Q_r} = \frac{\Delta x \frac{\frac{W}{\Delta x}\Omega \sqrt{\frac{\Delta z}{\Delta x}}H^{5/3}}{n}}{\frac{W\sqrt{\frac{\Delta z}{L}}H^{5/3}}{n}} = \Omega\sqrt{\frac{L}{\Delta x}} = \Omega\sqrt{\Omega} = \Omega^{3/2} \qquad (A.5)$$



**Appendix B: Input files for cases considered in this manuscript**
**Conceptual verification cases: straight and meandering channels**
mmax        = 11
nmax       = 26
dx         = 200
dy         = 200
x0         = -1000
y0         = 0
rotation    = 0
latitude    = 0
crsgeo     = 0
tref        = 20190101 000000
tstart     = 20190101 000000
tstop      = 20190103 000000
tspinup    = 60
dtmapout   = 86400
dthisout   = 600
dtmaxout   = 3600
dtwnd      = 1800
alpha      = 0.5
theta      = 0.95
huthresh   = 0.005
manning    = 0.02
manning_land  = 0.02
manning_sea   = 0.02
rgh_lev_land  = 0
zsini      = 1
qinf       = 0
rhoa       = 1.25
rhow       = 1024
dtmax      = 999
maxlev     = 999
bndtype    = 1
advection   = 0
baro       = 0
pavbnd     = 0
gapres     = 101200
advlim     = 5
stopdepth   = 100
depfile    = sfincs.dep
mskfile    = sfincs.msk
indexfile   = sfincs.ind
bndfile    = sfincs.bnd
bzsfile    = sfincs.bzs
srcfile    = sfincs.src
disfile    = sfincs.dis
sbgfile    = sfincs.sbg
obsfile    = sfincs.obs



```
crsfile       = sfincs.crs
manningfile   = sfincs.manning
inputformat   = bin
outputformat  = net
cdnrb         = 3
cdwnd         = 0  28  50
cdval         = 0.001    0.0025     0.0015
hmaxfile      = hmax.txt
zsfile        = zs.txt
dtout         = 3600
dttype        = min
storevelocity = 1
storevel      = 1
```

**630    Tidal propagation St. Johns River**

```
mmax          = 2720
nmax          = 5520
dx            = 25
dy            = 25
x0            = 459437.0
y0            = 3375791.0
rotation      = -164.0
epsg          = 32617
latitude      = 0.0
tref          = 20180901 000000
tstart        = 20180901 000000
tstop         = 20180931 000000
tspinup       = 60.0
dtout         = 86400
dthisout      = 600.0
dtrstout      = 0.0
dtmaxout      = 99999999999
trstout       = -999.0
dtwnd         = 1800.0
alpha         = 0.5
theta         = 1.0
huthresh      = 0.01
manning       = 0.04
manning_land  = 0.04
manning_sea   = 0.02
rgh_lev_land  = 0.0
zsini         = 0.0
qinf          = 0.0
rhoa          = 1.25
rhow          = 1024.0
dtmax         = 60.0
advection     = 2
baro          = 0
pavbnd        = 0
```



```
gapres         = 101200.0
stopdepth      = 100.0
crsgeo         = 0
btfilter       = 60.0
viscosity      = 1
depfile        = sfincs.dep
mskfile        = sfincs.msk
indexfile      = sfincs.ind
bndfile        = ..//..//setup//sfincs.bnd
bzsfile        = ..//..//setup//sfincs.bzs
sbgfile        = sfincs_subgrid.nc
obsfile        = ..//..//setup//noaa_xtide_v4_added_debug_points.obs
inputformat    = bin
outputformat   = net
cdnrb          = 3
cdwnd          = 0.0 28.0 50.0
cdval          = 0.001 0.0025 0.0015
```

**682    Conceptual verification cases: straight and meandering channels**

```
mmax           = 2632
nmax           = 1555
dx             = 25
dy             = 25
x0             = 243943.538
y0             = 3279280.3807
rotation       = 0
epsg           = 32615
tref           = 20170825 000000
tstart         = 20170825 000000
tstop          = 20170831 000000
dtout          = 86400
dthisout       = 600
dtmaxout       = 518400
dtwnd          = 600
alpha          = 0.5
theta          = 1
huthresh       = 0.05
rgh_lev_land   = 0
zsini          = 0
qinf           = 0
rhoa           = 1.25
rhow           = 1000
advection      = 1
stopdepth      = 9999
depfile        = sfincs.dep
mskfile        = sfincs.msk
indexfile      = sfincs.ind
bndfile        = sfincs.bnd
bzsfile        = sfincs.bzs
```



```
srcfile         = sfincs.src
disfile         = sfincs.dis
sbgfile         = sfincs_subgrid.nc
amprfile        = Observations_Interpolate_600x600_halfhour_test.amr
obsfile         = sfincs.obs
inputformat     = bin
outputformat    = net
cd_nr           = 0
geomskfile      = sfincs.gms
hmaxfile        = hmax.dat
hmaxgeofile     = hmaxgeo.dat
zsfile          = zs.dat
vmaxfile        = vmax.dat
qinffile        = qinf_constanttime_spatialvary
storevel        = 1
```



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
