# Peer review of "Subgrid corrections for the linear inertial equations of a compound flood model – a case study using SFINCS 2.1.1 Dollerup release"

_EGUsphere, 2024_

## Referee Comment (RC1)

**REVIEW, "A subgrid method for the linear inertial equations of a compound flood model"**

ORMONDT, LEIJNSE, GOEDE, NEDEROFF, DONGEREN

**General Comments**

In the manuscript, "A subgrid method for the linear inertial equations of a compound flood model," the authors describe a new subgrid model for use in improving the accuracy and efficiency of the coastal flooding model SFINCS. They found that the addition of subgrid corrections significantly improved model skill when compared to not using subgrid corrections, and only added minor computational expense. However, this added expense was insignificant when compared to the reduced computational cost of running on coarsened numerical grids. The authors also presented potential solutions to some of the problems often associated with running subgrid models on coarsened computational grids.

It is this reviewer's recommendation that the manuscript be accepted with minor revisions.

This manuscript is well written, and the following comments are suggestions for improving and clarifying the work.

This reviewer has the following questions and suggestions for the authors:

**Specific Comments**

1. Line 41-44: Although some full physics models have higher computational expense, that does not necessarily limit their application. For example ADCIRC is used to predict flooding on ocean and global scale numerical meshes in real time. This reviewer is not sure it the computational expense is actually limiting, it simply requires more computing power. Please revise this statement.

2. Line 48-49: The authors state that reduced the reduced-complexity models 'solve only the essential terms in the momentum equations'. How do the authors define essential? Instead, this reviewer would recommend changing this statement to 'These models solve reduced forms of the momentum...'. In addition, the authors compare to 'conventional models' at the end of the sentence, please change 'conventional' to 'full complexity' so that this doesn't get confused with the non-subgrid SFINCS model further into the paper. This reviewer would also like it if the specific momentum terms that are left out of a model like SFINCS are given as an example here.

3. Line 75: This reviewer believes V. Casulli's 2019 paper "Computational grid, subgrid, and pixels" was the first to introduce cell clones into a subgrid model. Consider citing this as well as Begmohammadi et al (2021) throughout the paper.

4. Line 197: The use of 20 levels between zmin and zmax of a subgrid area would likely work well for locations where there is only a few meters of difference in zmin and zmax. How many levels would the authors recommend for a much larger difference in zmin and zmax similar to what you might find if the subgrid area straddled a deep channel with a high bluff?

5. Lines 384-385: This reviewer would like to see a larger discussion on the computational expense of the subgrid code. What are the file sizes of the lookup tables? What file

type is used? NetCDF? Does the computational cost scale linearly with the grid/file size?

6. Lines 475: Why is there a range for the computational speed up from the 100 m to 25 m grid? Also in Table 2, why is the run time of the subgrid 50m less than the regular 50m? This seems inconsistent with the discussion. I would be nice to see the conputational cost increase added to a table.

**Technical Comments**

- Line 29-30: Suggest removing 'Furthermore, flood... save lives.'
- Remove empty box in Equations 3, 4, 17, 18, 22, 24 and Line 242.
- Make the averaging brackets in Equations 10, 11, and 13 larger to encompass entire term.
- Figure 1, 3, 5: consider changing color maps from rainbow.
- Line 241, 243 and 244: Need to add subscripts to 'zu', 'nu', and 'phiu'.
- Lines 361-369: For some reason the pdf made these lines bold with different vertical line spacing.
- Line 345: The authors mention the 25, 50, 100, 200, and 500 m test cases, but not the 1000 m test case listed in Table 1. Consider removing the 1000 m from the table.
- Line 370: Formatting error where line starts with 'Table 1).'.
- Lines 374-380: Formatting issues.
- Line 381: Formatting error where line starts with 'Table 1).'.
- Line 413: Add 'the' between 'on' and 'Sebastian'.
- Line 421: Could the authors please give the equation for the NSE in the discussion for reference?
- Figure 10: Could the authors use different markers/colors for the USGS stream gauges and HWMs on the map. The current ones are hard to see.
- Line 489: Again, I would consider citing Casulli 2019.
- Line 490: What do the authors mean by 'snapped'?
- Line 512: Add 's' to 'subgrid correction'.
- Line 513: Consider adding comma after 'channel'.
- Line 521: Consider changing 'subgrid corrections' benefits' to 'benefits of subgrid corrections'.

---

## Referee Comment (RC2)

This paper primarily focuses on the application of a subgrid method to simulate compound flooding scenarios, a critical issue in coastal systems that has gained increasing attention in recent years. The method is implemented using the SFINCS model and validated through several examples. While the manuscript is generally well-written and clear, it lacks some important details that could enhance its comprehensiveness.

1- There are now several subgrid (SG) models available in the field, such as CoasToRM and the latest version of HEC-RAS. The authors should cite these models and discuss the key differences, highlighting the advantages of their approach in comparison to these existing models.

References:

"Begmohammadi, Amirhosein, Damrongsak Wirasaet, Ning Lin, J. Casey Dietrich, Diogo Bolster, and Andrew B. Kennedy. "Subgrid modeling for compound flooding in coastal systems." *Coastal Engineering Journal* (2024): 1-18."

"Brunner, G. "HEC-RAS River Analysis System Version 5.0—Hydraulic Reference Manual." *Hydrologic Engineering Center, Davis, California, US* (2016)."

2- Equation (2) shows the upscaled mass conservation equation with additional source terms, $S$.

How do the authors include infiltration in the model. I think more explanation about infiltration is needed since the vertical infiltration process during the first period of rainfall has more impact on large-scale flooding. What kind of infiltration model is used in subgrid SFINCS? For example, following infiltration model is proposed by Raws et al., 1992;

How do the authors incorporate infiltration into the model? Additional explanation on this aspect is necessary, as the vertical infiltration process during the initial phase of rainfall significantly influences large-scale flooding. What type of infiltration model is employed in the subgrid SFINCS? For instance, Raws et al. (1992) proposed a model that could be relevant here.

$$ f = k \left(1 + \frac{(\varphi - \theta)S}{F}\right) $$

where k is the vertical saturated hydraulic conductivity, $\varphi$ is the soil porosity, $\theta$ is the initial water volume content, S is the suction at the vertical wetting front and f is the cumulative infiltration depth.

3- In Equation (2), is the matrix always positive definite?
4- In Section 3: Conceptual Verification Cases—Straight and Meandering Channels, the authors present the meandering river example. They demonstrate that the discharge for 100m, 200m, and 500m subgrid resolutions is inaccurate. Two reasons are cited: that "the channel is effectively schematized as a straight channel with a length of 5000 m. This leads to an overestimation of the true water level slope and resulting in a wet average flux. Secondly, meanders inside a grid cell result in a larger wet fraction, which the model "interprets" as a wide channel, leading to further overestimation." I believe the authors may not have implemented this test case correctly.

Accurate bottom friction is essential for this scenario, which I do not think they accounted for. Please refer to the following papers:

Volp, N. D., Van Prooijen, B. C., & Stelling, G. S. (2013). A finite volume approach for shallow water flow accounting for high-resolution bathymetry and roughness data. Water Resources Research, 49(7), 4126–4135. https://doi.org/10.1002/wrcr.20324.

Kennedy, A. B., Wirasaet, D., Begmohammadi, A., Sherman, T., Bolster, D., & Dietrich, J. C. (2019). Subgrid theory for 756 storm surge modeling. Ocean Modelling, 144, 101491. https://doi.org/10.1016/j.ocemod.2019.101491.

In both papers, they consider this problem as a 1 dimensional channel (The grid they used is larger than the current study). They still get a very good result. Can authors explain the friction scheme used here? Can they make a comment if their friction is equivalent to these two papers?

5- In the Hurricane Harvey example, they mention that the high resolution 25 m model has a fair correlation with observation. Can you quantify that? What do you call fair correlation?

6- There is extensive High Water Mark (HWM) data available for this region from Hurricane Harvey. Would it be possible to compare these high water marks with the model simulations? This comparison could provide a clearer evaluation of the model's performance across different grid resolutions, including the subgrid approach.

7- This section could benefit from additional figures highlighting the difference between model runs that in/exclude rain/infiltration/river discharge input, to distinguish the importance of these drivers for the inland part.

8- Regarding this DEM, is river bathymetry (sufficiently) included in this dataset? Often it is not very accurate in lidar based DEMs, if not treated afterwards. If so, how might that affect the inland flooding results.

9- The SFINCS model can be run on a GPU. Does the subgrid version have the same capability?

10- Figure.9 and related descriptions: Is it possible that hourly rainfall intensity (i.e. hyetograph) is shown with time series of water surface elevation in Fig.7? I think it is helpful for understanding the relationship between the peak of water surface elevation and the precipitation

11- There are a lot of minor problems in writing and equations: for instance, line 241: zu

---

## Author Comment (AC1)

**Rebuttal letter manuscript "A subgrid method for the linear inertial equations of a compound flood model"**

Dear Editor, dear reviewers,

On 17 Jun 2024, we submitted the following manuscript to Geoscientific Model Development (GMD) titled: " A subgrid method for the linear inertial equations of a compound flood model" (https://doi.org/10.5194/egusphere-2024-1839). On 10 October 2024, we were informed that the discussion on EGUsphere was closed. In total, we received comments from two referees, one person from the community, and the chief editor, who all provided very positive feedback on the work done and valid suggestions. We would like to acknowledge their time and efforts, which have led to an improvement in the quality of our manuscript. Below you find a point-by-point reply to all specific questions and suggestions. Attached you also find the revised manuscript with the changes made to address the review comments tracked.

Kind regards,

Kees Nederhoff

**CEC1: Astrid Kerkweg, 05 Aug 2024**

Dear authors,

in my role as Executive editor of GMD, I would like to bring to your attention our Editorial version 1.2: https://www.geosci-model-dev.net/12/2215/2019/.

This highlights some requirements of papers published in GMD, which is also available on the GMD website in the 'Manuscript Types' section: http://www.geoscientific-model-development.net/submission/manuscript_types.html

In particular, please note that for your paper, the following requirements have not been met in the Discussions paper:

- "The main paper must give the model name and version number (or other unique identifier) in the title."
- "If the model development relates to a single model then the model name and the version number must be included in the title of the paper. If the main intention of an article is to make a general (i.e. model independent) statement about the usefulness of a new development, but the usefulness is shown with the help of one specific model, the model name and version number must be stated in the title. The title could have a form such as, "Title outlining amazing generic advance: a case study with Model XXX (version Y)"."

As you implemented your method into the SFINCS model, please add something like "a case study using SFINCS version x.y" to the title of your manuscript in your revised submission to GMD.

We want to thank the Executive Editor for pointing this out. In the revised manuscript, we changed the title and included the model name and version number (Line 2)

Yours, Astrid Kerkweg

**RC1: Anonymous Referee #1, 23 Aug 2024**

**General Comments**

In the manuscript, "A subgrid method for the linear inertial equations of a compound flood model," the authors describe a new subgrid model for use in improving the accuracy and efficiency of the coastal flooding model SFINCS. They found that the addition of subgrid corrections significantly improved model skill when compared to not using subgrid corrections, and only added minor computational expense. However, this added expense was insignificant when compared to the reduced computational cost of running on coarsened numerical grids. The authors also presented potential solutions to some of the problems often associated with
running subgrid models on coarsened computational grids.

It is this reviewer's recommendation that the manuscript be accepted with minor revisions.

This manuscript is well written, and the following comments are suggestions for improving and clarifying the work.

*We want to thank Anonymous Referee #1 for the constructive feedback. We addressed their points below.*

This reviewer has the following questions and suggestions for the authors:

**Specific Comments**

1. Line 41-44: Although some full physics models have higher computational expense, that does not necessarily limit their application. For example ADCIRC is used to predict flooding on ocean and global scale numerical meshes in real time. This reviewer is not sure it the computational expense is actually limiting, it simply requires more computing power. Please revise this statement.

*We agree with the reviewer and changed this statement in the revised manuscript (L35-47). In particular, we have revised the manuscript to clarify the computational demands and practical constraints of classical full-physics complexity models (e.g., ADCIRC, Delft3D-FLOW, MIKE, and SOBEK). The added text highlights that while these models provide highly detailed simulations, their high computational costs may limit their applicability in large-scale, high-resolution, or time-sensitive scenarios, especially when exploring flooding uncertainties through ensemble modeling.*

2. Line 48-49: The authors state that reduced the reduced-complexity models 'solve only the essential terms in the momentum equations'. How do the authors define essential? Instead, this reviewer would recommend changing this statement to 'These models solve reduced forms of the momentum…'. In addition, the authors compare to 'conventional models' at the end of the sentence, please change 'conventional' to 'full complexity' so that this doesn't get confused with the non-subgrid SFINCS model further into the paper. This reviewer would also like it if the specific momentum terms that are left out of a model like SFINCS are given as an example here.

*We changed this statement in the revised manuscript (L49-54). Specifically, we revised the statement to "reduced forms of the momentum equations". Additionally, we replaced "conventional models" with "full complexity models" to avoid any potential confusion with the non-subgrid version of SFINCS later in the paper.*

Regarding the reviewer's request for details on specific momentum terms, our focus in this manuscript is on introducing subgrid corrections rather than dissecting the omitted terms in reduced-complexity models. These simplifications are discussed in prior work, including Leijnse et al. (2021), and are addressed in Section 2.1 (L106-125) of the manuscript.

3. Line 75: This reviewer believes V. Casulli's 2019 paper "Computational grid, subgrid, and pixels" was the first to introduce cell clones into a subgrid model. Consider citing this as well as Begmohammadi et al (2021) throughout the paper.

We agree with the reviewer and added these references through the revised manuscript (L84 and L530).

4. Line 197: The use of 20 levels between zmin and zmax of a subgrid area would likely work well for locations where there is only a few meters of difference in zmin and zmax. How many levels would the authors recommend for a much larger difference in zmin and zmax similar to what you might find if the subgrid area straddled a deep channel with a high bluff?

In our sensitivity tests, we observed that 20 levels generally performed well across moderate elevation ranges. For areas with significantly larger elevation differences, such as deep channels and high bluffs, a higher number of levels might be necessary to capture the full complexity of the terrain. However, the optimal number of levels will depend on site-specific factors, including the steepness of gradients and desired model resolution. We added these considerations in the revised manuscript (L208-L215, L490-506).

5. Lines 384-385: This reviewer would like to see a larger discussion on the computational expense of the subgrid code. What are the file sizes of the lookup tables? What file type is used? NetCDF? Does the computational cost scale linearly with the grid/file size?

We have expanded the discussion on computational costs in the revised manuscript (L490-502). We have also added information regarding the file sizes of the subgrid lookup tables and the file format used (L502-506). Specifically, the subgrid tables are stored in NetCDF format, which is commonly used in hydrodynamic modeling. Regarding file sizes, for example, in the 200-meter resolution Harvey case, the subgrid file size was 343 MB, while for the 200-meter Jacksonville case, it was 65 MB. This demonstrates how the file size scales nearly linearly with the number of active cells and the number of discrete bins. This scaling behavior also affects the computational costs, which increase proportionally with finer binning and larger file sizes.

6. Lines 475: Why is there a range for the computational speed up from the 100 m to 25 m grid? Also in Table 2, why is the run time of the subgrid 50m less than the regular 50m? This seems inconsistent with the discussion. I would be nice to see the conputational cost increase added to a table.

We have revisited the timing of the computational costs by rerunning the model with the current release and repeating the timing three times to ensure more reliable results. Previously, we found that the run times were potentially "contaminated" as the tests were conducted on a Windows-based machine. In our current findings, we do not observe that the subgrid version is faster than the regular version on the same grid resolution. Instead, we consistently see an increase in computational costs when adding subgrid tables. Additionally, the computational cost increase was higher in the Harvey case, where we used 100 bins instead of the more typical 20 bins (L495-502). This explains the variation in computational expenses across different scenarios. We have updated

the discussion to reflect these findings and ensured that the computational cost increase is clearly outlined in the revised manuscript.

**Technical Comments**

- Line 29-30: Suggest removing 'Furthermore, flood... save lives.'

We followed the suggestion of RC1 and removed this statement.

- Remove empty box in Equations 3, 4, 17, 18, 22, 24 and Line 242.

We removed the empty boxes at the equations.

- Make the averaging brackets in Equations 10, 11, and 13 larger to encompass entireterm.

We appreciate this suggestion and agree that larger averaging brackets would enhance clarity in Equations 10, 11, and 13. While we were unable to implement this change directly in Microsoft Word, we kindly request that the typesetting team address this adjustment during the final production stage to ensure the brackets properly encompass each term.

- Figure 1, 3, 5: consider changing color maps from rainbow.

We are aware of the limitations typically associated with rainbow color maps, particularly concerning perceptual uniformity and accessibility for color-blind readers. However, after careful consideration, we feel that the rainbow color map effectively distinguishes the range of values pertinent to our analysis and communicates spatial variations in a clear and impactful manner for the target audience.

- Line 241, 243 and 244: Need to add subscripts to 'zu', 'nu', and 'phiu'.

Thank you for pointing this out. We have added the necessary subscripts to 'zu,' 'nu,' and 'phiu' in the revised manuscript.

- Lines 361-369: For some reason the pdf made these lines bold with different vertical line spacing.

We have removed these lines in bold and with a different line spacing so that the entire manuscript is consistent.

- Line 345: The authors mention the 25, 50, 100, 200, and 500 m test cases, but not the 1000 m test case listed in Table 1. Consider removing the 1000 m from the table.

We have removed the 1000m test case in the table.

- Line 370: Formatting error where line starts with 'Table 1).'.
- Lines 374-380: Formatting issues.
- Line 381: Formatting error where line starts with 'Table 1).'.

We addressed these formatting issues in the revised manuscript (L381-411).

- Line 413: Add 'the' between 'on' and 'Sebastian'.

Added 'the' in the revised manuscript (L425).

- Line 421: Could the authors please give the equation for the NSE in the discussion for reference?

We added the equation for the NSE as a footnote.

- Figure 10: Could the authors use different markers/colors for the USGS stream gauges and HWMs on the map. The current ones are hard to see.

We revised Figures 8 and 10 and made the USGS stream gauges visible as red stars and the HWM as yellow solid circles.

- Line 489: Again, I would consider citing Casulli 2019.

We added the Casulli reference here too (L530)

- Line 490: What do the authors mean by 'snapped'?

"Snapped" refers to aligning or fitting the subgrid features (like weirs) precisely to the computational grid. We rephrased this sentence and used 'aligned with' now (L532).

- Line 512: Add 's' to 'subgrid correction'.

We added the 's' here and made it "subgrid corrections".

- Line 513: Consider adding comma after 'channel'.

We added a ',' here and made it "For the meandering channel, differences "

- Line 521: Consider changing 'subgrid corrections' benefits' to 'benefits of subgrid corrections'.

Followed the suggestion of the reviewer and made the sentence "Real-world application cases further validated the benefits of subgrid correction".

**CC1, Jingming Hou, 27 Aug 2024**

**General Overview:**

The paper presents a novel approach integrating subgrid corrections into the SFINCS model to enhance the accuracy of flood simulations while reducing computational costs. The study is well-motivated, considering the importance of accurate and efficient flood risk assessments, especially in coastal areas prone to compound flooding. The methodology, results, and validation against both conceptual and real-world cases are robust and demonstrate the potential benefits of the proposed approach.

We want to thank dr. Jingming Hou for the constructive feedback. We addressed their points below.

**Major Comments:**

1. Clarity of the Methodology:

The paper's description of the subgrid approach is detailed, but certain sections, particularly the mathematical derivations, may benefit from additional clarification or simplification. Equations such as (7) and (11) are critical to the paper's argument but might be difficult for non-specialists to follow. Including a more intuitive explanation or visual aids to complement these equations could make the content more accessible.

We appreciate the suggestion and recognize the importance of making complex methodologies accessible. However, the mathematical derivations in sections containing Equations (7) and (11) are primarily intended for a specialized audience familiar with advanced hydrodynamic modeling techniques and this type of equations. The purpose of these sections is to provide a rigorous theoretical background for the approach, which may not be directly accessible to non-specialists. We hope that the main results and visualizations in the later sections offer an intuitive understanding of the subgrid approach's implications. Consequently, no changes were made to the manuscript in response to this comment.

The discussion of the limitations of the method, particularly regarding the handling of unresolved meanders and flow-blocking features, should be expanded. It is important to discuss how these issues might affect the model's performance in different real-world scenarios and suggest possible future research directions or improvements.

We discussed the limitations related to unresolved meanders and flow-blocking features in the current manuscript (L529-536). Specifically, we address how insufficient river bathymetry and unresolved meanders can affect model performance, particularly for riverine flooding. We also note that the subgrid weir formulation used in SFINCS presents challenges in capturing flow-blocking features, and we refer to ongoing research efforts aimed at improving this aspect. However, we acknowledge the importance of expanding this discussion, and we have therefore expanded the revised manuscript text to better explain how these limitations could impact model performance across different real-world scenarios (L538-545).

Validation and Case Studies:

The validation of the subgrid method using both conceptual and real-world cases is a strong point. However, the selection of cases could be more diverse. For instance, including a case study from a different geographical area or a different type of flooding (e.g., urban pluvial flooding in a densely populated area) could demonstrate the broader applicability of the method.

We appreciate the reviewer's suggestion to include a more diverse selection of case studies. However, the primary focus of this work is to introduce and validate the subgrid corrections for the

Linear Inertial Equations (LIE) within the SFINCS model framework. Our aim was to assess the performance of these corrections across different model schematizations already established in the literature. We selected three case studies—conceptual, riverine, and urban flooding—which we believe cover a broad spectrum of hydrodynamic conditions and types of flooding.

We feel that these cases provide valuable insights into the performance of the subgrid corrections across different settings and are sufficient to demonstrate the applicability of the method. Expanding to additional geographical areas or other types of flooding is indeed an important next step, but we consider it beyond the scope of this manuscript. We believe that the current selection of cases adequately showcases the method's potential, especially considering that the subgrid corrections were designed to be versatile and applicable in various contexts.

We look forward to future studies that can build on this foundation by exploring subgrid corrections in even more diverse environments. Consequently, no changes were made to the manuscript in response to this comment.

The performance metrics provided (e.g., RMSE, NSE) are appropriate, but a deeper statistical analysis comparing the subgrid approach to other established methods (beyond just the regular SFINCS model) could strengthen the paper's claims.
We appreciate the reviewer's suggestion to include a more detailed statistical comparison of the subgrid approach with other established methods. In this paper, our primary goal was to introduce and validate the subgrid corrections for the Linear Inertial Equations (LIE) within the SFINCS model. We believe that the current performance metrics, such as RMSE and NSE, are appropriate for demonstrating the improvement achieved by the subgrid approach over the regular SFINCS model.

While we agree that a deeper statistical comparison with other established methods could further strengthen our claims, we consider it beyond the scope of this study. The focus here is on the subgrid corrections, and comparing them against other approaches would require a more extensive analysis that might dilute the primary contribution of this work.

We see this as an exciting opportunity for future research, where the subgrid approach can be evaluated against a wider range of models in different contexts. We believe the current analysis is sufficient to demonstrate the efficacy of the subgrid corrections and their potential for broad applications.

Computational Efficiency:
The paper mentions that integrating subgrid corrections increases computational costs by 44 to 129%, which is significant. A more detailed discussion on the trade-off between computational efficiency and accuracy would be beneficial. For instance, providing guidance on when it might be preferable to use the subgrid method despite the increased cost would help practitioners.

Additionally, discussing how these computational costs compare to those of alternative models or approaches would provide readers with a better context for evaluating the subgrid method's efficiency.
We have expanded the discussion on computational costs in the revised manuscript (L490-506). We agree that discussing how these computational costs compare to alternative models or approaches could provide valuable context. In this case, the subgrid method does introduce additional computational expense, but it offers improved accuracy and the ability to use coarser grids while

maintaining performance (see also discussion L490-491). Therefore, in practical terms, it is always worthwhile to use subgrid corrections.

Figures and Tables:
Figures 7 and 10, which illustrate the model results for the St. Johns River and Hurricane Harvey cases, are informative but could be enhanced by including additional comparative plots. For example, showing the differences in predicted flood extents or water levels between the regular and subgrid models in a side-by-side comparison could provide a clearer visual demonstration of the subgrid method's benefits.
We thank the reviewer for their feedback and the suggestion to enhance the visual comparison between the regular and subgrid models. We would like to highlight that Figure 10 already provides a detailed comparison of the flood inundation for different SFINCS model configurations, including both regular and subgrid approaches.

Figure 10 shows the modeled flood inundation in the midstream portion of Brays Bayou for four different SFINCS model options:
- Panel A displays the results using the regular 25m resolution.
- Panel B shows the regular 100m resolution.
- Panel C presents the subgrid 100m model, but with downscaled water depths applied as a post-processing step.
- Panel D shows the subgrid 100m model without the downscaling.

This figure provides a clear side-by-side comparison between different resolutions and the subgrid methods, demonstrating the benefits of subgrid corrections. Specifically, it highlights the ability of the subgrid approach to improve model accuracy at coarser grid resolutions (100m), showing similar performance to the finer grid resolution (25m) without subgrid corrections. The differences in predicted flood extents are visually apparent in the inundation maps.

We believe this figure provides a detailed visual demonstration of the subgrid method's benefits. However, we acknowledge that further comparative plots could always add value, and we will consider such additions in future work.

The tables summarizing the model's performance metrics (e.g., Table 1 and 2) are useful, but adding a column that explicitly shows the percentage improvement (or decline) in performance when using the subgrid method compared to the regular SFINCS model would help highlight the method's effectiveness.
We appreciate the reviewer's suggestion to include a more explicit comparison of the performance metrics between the subgrid method and the regular SFINCS model. In response, we have added a column to both Table 1 and Table 2 that shows the percentage error relative to the regular_25m model. This additional column highlights the percentage improvement (or decline) in performance when using the subgrid method compared to the regular SFINCS model.

We believe this modification provides clearer insight into the effectiveness of the subgrid corrections and makes the comparison between the different model configurations more transparent and informative.

**Minor Comments:**

**1. Terminology:**

Ensure consistency in the use of terminology throughout the manuscript. For instance, terms like "subgrid corrections," "subgrid method," and "subgrid-enabled" should be clearly defined and used consistently to avoid confusion.

Thank you for your suggestion. In response, we have revised the manuscript to only mention either 'subgrid corrections' or 'the subgrid version of SFINCS', including the title.

**2. Typographical Errors:**

There are a few minor typographical errors in the manuscript. For example, the equation numbering sometimes skips or duplicates, which can confuse the reader. Ensure that all equations are numbered sequentially and referenced correctly in the text.

A thorough proofread to correct any such errors is recommended before resubmission.

Thank you for highlighting this. We have thoroughly proofread the manuscript and corrected all typographical errors, including ensuring that equation numbering is sequential and consistently referenced throughout the text.

**3. Conclusion Section:**

The conclusion effectively summarizes the paper's key findings, but it could be strengthened by adding a few sentences on the potential future applications of the subgrid method. For example, discussing how this method could be adapted or expanded to other types of hydrodynamic models or different environmental conditions would provide a more forward-looking perspective.

Thank you for your suggestion. In response, we have revised the conclusion to add potential future applications of the subgrid method. We highlight how this method could be adapted or expanded to other types of hydrodynamic models and different environmental conditions. This addition provides a more forward-looking perspective and emphasizes the broader relevance and applicability of the approach (L573-580).

**Recommendation:**

The paper presents significant advancements in hydrodynamic modeling, particularly in the efficient simulation of compound floods. With the suggested revisions, particularly regarding the clarity of the methodology and the expansion of case studies, I believe the manuscript would make a valuable contribution to the field and recommend it for publication after minor revisions.

These comments should provide the authors with constructive feedback to refine their manuscript and address any potential concerns that might arise during peer review.

We appreciate your positive feedback on our manuscript and your recognition of the advancements in hydrodynamic modeling presented in the paper. We are grateful for your thoughtful suggestions regarding the methodology's clarity and the expansion of case studies. In response, we have addressed all of your comments and made the necessary revisions to improve the clarity and scope of the manuscript.

**RC2: Anonymous Referee #2, 10 Sep 2024**

This paper primarily focuses on the application of a subgrid method to simulate compound flooding scenarios, a critical issue in coastal systems that has gained increasing attention in recent years. The method is implemented using the SFINCS model and validated through several examples. While the manuscript is generally well-written and clear, it lacks some important details that could enhance its comprehensiveness.

1. There are now several subgrid (SG) models available in the field, such as CoasToRM and the latest version of HEC-RAS. The authors should cite these models and discuss the key differences, highlighting the advantages of their approach in comparison to these existing models.

Thank you for your valuable suggestion. In response, we have now cited additional subgrid models, including CoaSToRM and HEC-RAS, which also incorporate subgrid approaches (L77-81). We have also discussed the key differences between these and other models referenced in the introduction and the subgrid corrections for Linear Inertial Equations (LIE) introduced in this paper throughout the manuscript (L102-107).

References:
"Begmohammadi, Amirhosein, Damrongsak Wirasaet, Ning Lin, J. Casey Dietrich, Diogo Bolster, and Andrew B. Kennedy. "Subgrid modeling for compound flooding in coastal systems." Coastal Engineering Journal (2024): 1-18."
"Brunner, G. "HEC-RAS River Analysis System Version 5.0—Hydraulic Reference Manual." Hydrologic Engineering Center, Davis, California, US (2016)."

2. Equation (2) shows the upscaled mass conservation equation with additional source terms, S.

How do the authors include infiltration in the model. I think more explanation about infiltration is needed since the vertical infiltration process during the first period of rainfall has more impact on large-scale flooding. What kind of infiltration model is used in subgrid SFINCS? For example, following infiltration model is proposed by Raws et al., 1992;

How do the authors incorporate infiltration into the model? Additional explanation on this aspect is necessary, as the vertical infiltration process during the initial phase of rainfall significantly influences large-scale flooding. What type of infiltration model is employed in the subgrid SFINCS? For instance, Raws et al. (1992) proposed a model that could be relevant here.

$$f = k \left(1 + \frac{(\varphi - \theta)S}{F}\right)$$

where k is the vertical saturated hydraulic conductivity, φ is the soil porosity, θ is the initial water volume content, S is the suction at the vertical wetting front and f is the cumulative infiltration depth. In the revised manuscript, we have added a more detailed explanation of how infiltration is handled. In particular, in SFINCS, infiltration can be modeled using either constant in-time infiltration rates or

more sophisticated, empirically based rainfall-runoff models (L128-132). Empirical methods such as the Curve Number method, the Green-Ampt method, and the Horton infiltration are widely used in hydrological modeling to simulate vertical infiltration processes during rainfall events and are suited to represent different soil conditions and rainfall intensities.

The Raws et al. (1992) proposed model appears to be similar to the basic form of the Green-Ampt equation which is expressed as follows: $f(t) = K(1+ \text{delta\_theta} (\text{sigma} + h0) / F(t) )$.

Furthermore, we have added additional discussion on how infiltration rates are computed on the computational grid in the subgrid SFINCS model. This revision acknowledges that the current approach does not incorporate higher-resolution information for estimating infiltration rates, which may reduce accuracy in areas with significant variability (L516-520).

3. In Equation (2), is the matrix always positive definite?

No, the matrix in Equation (2) is not always positive definite, as water levels can indeed become negative under certain conditions. We address this by limiting the flow when a cell becomes dry, which ensures stability in areas where water depth approaches zero or becomes negative. This approach helps manage flow in scenarios with varying water levels, maintaining model robustness even in cases of drying cells.

4. In Section 3: Conceptual Verification Cases—Straight and Meandering Channels, the authors present the meandering river example. They demonstrate that the discharge for 100m, 200m, and 500m subgrid resolutions is inaccurate. Two reasons are cited: that "the channel is effectively schematized as a straight channel with a length of 5000 m. This leads to an overestimation of the true water level slope and resulting in a wet average flux. Secondly, meanders inside a grid cell result in a larger wet fraction, which the model "interprets" as a wide channel, leading to further overestimation." I believe the authors may not have implemented this test case correctly. Accurate bottom friction is essential for this scenario, which I do not think they accounted for. Please refer to the following papers:

Volp, N. D., Van Prooijen, B. C., & Stelling, G. S. (2013). A finite volume approach for shallow water flow accounting for high-resolution bathymetry and roughness data. Water Resources Research, 49(7), 4126– 4135. https://doi.org/10.1002/wrcr.20324.

Kennedy, A. B., Wirasaet, D., Begmohammadi, A., Sherman, T., Bolster, D., & Dietrich, J. C. (2019). Subgrid theory for 756 storm surge modeling. Ocean Modelling, 144, 101491. https://doi.org/10.1016/j.ocemod.2019.101491.

In both papers, they consider this problem as a 1 dimensional channel (The grid they used is larger than the current study). They still get a very good result. Can authors explain the friction scheme used here? Can they make a comment if their friction is equivalent to these two papers?

In our study, for the straight channel case, we achieved results consistent with those presented by Volp et al. (2013) and Kennedy et al. (2019). Specifically, for the coarsest subgrid resolutions, our results match the high-resolution regular model exactly, as reported in these studies. However, for the meandering channel, we observed an increase in errors with larger grid sizes, which is also similar to the findings in the mentioned papers.

In this paper, we intentionally pushed the case to more extreme conditions by using a coarser discretization to capture the meandering, which led to an overestimation of the discharge due to the channel's geometry and grid resolution. We believe this more extreme setup provides additional insight into the limitations of the method under such conditions.

In SFINCS, bottom friction is computed using both x- and y-fluxes, which takes into account the subgrid corrections for both the bathymetry and the roughness elements at the subgrid level. While our approach to friction is similar to the work of Volp et al. (2013) and Kennedy et al. (2019), there may be subtle differences in how the friction and roughness data are handled, particularly in the representation of meanders and the subgrid variability.

Moreover, we believe that for practical cases the concept of sinuosity is more important than frictional effects. In particular, when using a computational grid that does not resolve the river meanders, the presented subgrid corrections may overestimate discharges by more than a factor of 5 (see for more background information the discussion L538-545)

5. In the Hurricane Harvey example, they mention that the high resolution 25 m model has a fair correlation with observation. Can you quantify that? What do you call fair correlation?

We have removed the subjective term "fair correlation" from the manuscript. We have instead quantified the model's performance by reporting the specific error (73 cm) when comparing the model results to observed data across the study area. This provides a clearer and more objective assessment of the model's accuracy (L414-416).

6. There is extensive High Water Mark (HWM) data available for *this* region from Hurricane Harvey. Would it be possible to compare these high water marks with the model simulations? This comparison could provide a clearer evaluation of the model's performance across different grid resolutions, including the subgrid approach.

Please note that the manuscript already includes a comparison of 115 high water mark (HWM) locations, following the analysis performed by Sebastian et al. (2021). These results are presented in Table 2, where the model's performance is evaluated across different grid resolutions, including the subgrid approach. We believe this comparison already provides a thorough assessment of the model's accuracy in relation to the available HWM data.

7. This section could benefit from additional figures highlighting the difference between model runs that in/exclude rain/infiltration/river discharge input, to distinguish the importance of these drivers for the inland part.

We appreciate the reviewer's suggestion to include additional figures highlighting the differences between model runs that include/exclude rain, infiltration, and river discharge inputs. This comparison of importance forcing was the subject of an MSc thesis that can be found here https://repository.tudelft.nl/record/uuid:57b9e495-0c90-4cf5-ab22-e169fb908ac1. However, the primary focus of the present paper is to introduce the subgrid corrections for the Linear Inertial Equations (LIE) and to test these corrections on existing studies published in the literature. While distinguishing the importance of different drivers such as rain, infiltration, and river discharge is valuable, we believe this falls outside the scope of our current work.

8. Regarding this DEM, is river bathymetry (sufficiently) included in this dataset? Often it is not very accurate in lidar based DEMs, if not treated afterwards. If so, how might that affect the inland flooding results.

Thank you for pointing this out. We agree that this is an important consideration, and we have now added this point to the revised discussion. As noted, river bathymetry is often insufficiently represented in combined topo-bathymetry datasets, especially when using LiDAR-based DEMs. This lack of accurate bathymetric data can affect the performance of hydrodynamic models, particularly in riverine flooding scenarios. In the cases presented here, no specific adjustments were made to improve river bathymetry, and the models were simply run at various resolutions with and without subgrid corrections. As a result, the inland flooding results may be influenced by these limitations. See also the revised manuscript L522-527.

9.   The SFINCS model can be run on a GPU. Does the subgrid version have the same capability?
Yes, the subgrid corrections are compatible with both the CPU and GPU versions of the SFINCS model. However, for this study, we focused on the implementation and testing of the subgrid corrections using the CPU version, primarily due to the availability of these computational resources. Future work could explore the performance benefits of running the subgrid version on a GPU.

10.   Figure.9 and related descriptions: Is it possible that hourly rainfall intensity (i.e. hyetograph) is shown with time series of water surface elevation in Fig.7? I think it is helpful for understanding the relationship between the peak of water surface elevation and the precipitation
Yes, we added the hourly rainfall intensity (i.e. hyetograph) to Figure 9 in the revised manuscript.

11.   There are a lot of minor problems in writing and equations: for instance, line 241: zu
We have carefully reviewed the manuscript and revised all instances of minor writing and equation issues, including the error on line 241 where "zu" was mentioned. We have ensured that the writing is clear and consistent and that all equations are correctly formatted and referenced.